# The control of anthropogenic emissions contributed to 80% of the decrease in PM$_{2.5}$ concentrations in Beijing from 2013 to 2017

Ziyue Chen[1,2], Danlu Chen[1], Mei-Po Kwan[3,4], Bin Chen[5] , Bingbo Gao[6*], Yan Zhuang[1], Ruiyuan Li[1], Bing Xu[7*]

[1]College of Global and Earth System Science,Beijing Normal University,19 Xinjiekou Street,Haidian,Beijing 100875,China.

[2]Joint Center for Global Change Studies, Beijing 100875,China.

[3]Department of Geography and Geographic Information Science, University of Illinois at Urbana-Champaign, Urbana, IL 61801, USA.

[4]Department of Human Geography and Spatial Planning, Utrecht University, 3584 CB Utrecht, The Netherlands

[5]Department of Land, Air and Water Resources, University of California, Davis, CA 95616, USA

[6]College of Land Science and Technology, China Agriculture University, Tsinghua East Road, Haidian District, 100083, China.

[7]Ministry of Education Key Laboratory for Earth System Modeling, Department of Earth System Science, Tsinghua University, Beijing 100084,China

[*]To whom correspondence should be addressed. Email: gaobb@lreis.ac.cn or bingxu@tsinghua.edu.cn

## Abstract

With the completion of the Beijing Five-year Clean Air Action Plan by the end of 2017, the annual mean PM$_{2.5}$ concentration in Beijing dropped dramatically to 58.0 μg/m$^3$ in 2017 from 89.5 μg/m$^3$ in 2013. However, controversies exist to argue that favorable meteorological conditions in 2017 were the major driver for such rapid decrease in PM$_{2.5}$ concentrations. To comprehensively evaluate this five-year plan, we employed Kolmogorov-Zurbenko (KZ) filter and WRF-CMAQ to quantify the relative contribution of meteorological conditions and the control of anthropogenic emissions to PM$_{2.5}$ reduction in Beijing from 2013 to 2017. For these five years, the relative contribution of emission-reduction to the decrease of PM$_{2.5}$ concentrations calculated by KZ filtering and WRF-CMAQ was 80.6% and 78.6% respectively. KZ filtering suggested that short-term variations of meteorological and emission conditions contributed majorly to rapid changes of PM$_{2.5}$ concentrations in Beijing.

WRF-CMAQ revealed that the relative contribution of local and regional
emission-reduction to $PM_{2.5}$ decrease in Beijing was 53.7% and 24.9% respectively.
For local emission-reduction measures, the regulation of coal boilers, increasing use
of clean fuels for residential use and industrial restructuring contributed to 20.1 %,
17.4% and 10.8% of $PM_{2.5}$ reduction respectively. Both models suggested that the
control of anthropogenic emissions accounted for around 80% of the $PM_{2.5}$ reduction
in Beijing, indicating that emission-reduction was crucial for air quality enhancement
in Beijing from 2013 to 2017. Consequently, such long-term air quality clean plan
should be continued in the following years to further reduce $PM_{2.5}$ concentrations in
Beijing.
**Keywords: $PM_{2.5}$, anthropogenic emissions, meteorological conditions,**
**Kolmogorov-Zurbenko (KZ) filtering, WRF-CMAQ**

## 1 Introduction

In January 2013, persistent haze episodes occurred in Beijing, during which the highest hourly $PM_{2.5}$ concentration once reached 886 μg/m$^3$, a historic high record. High-concentration $PM_{2.5}$ led to long-lasting black and thick fogs, which not only significantly influenced people's daily life (low-visibility induced traffic jam), but also posed a severe threat to public health (Brunekreef et al., 2002; Dominici et al., 2014; Nel et al., 2005; Zhang et al., 2012; Qiao et al., 2014). Since then, severe haze episodes have frequently been observed in Beijing and other regions across China (Chan et al., 2008; Huang, R., et al., 2014; Guo et al.,2014; Zheng et al.,2015), and $PM_{2.5}$ pollution has become one of the most concerned environmental issues in China. Consequently, a national network for monitoring hourly $PM_{2.5}$ concentrations has been established gradually, including 35 ground observation stations in Beijing, which provide important support for better understanding and managing $PM_{2.5}$ concentrations. To effectively mitigate $PM_{2.5}$ pollution, Beijing Municipal Government released "Beijing Five-year Clean Air Action Plan (2013-2017)" with a series of long-term emission-reduction measures, including shutting down heavily polluting factories, restricting traffic emissions and replacing coal fuels with clean energies, and "Heavy Air Pollution Contingency Plan" with a series of contingent emission-reduction measures during heavy pollution episodes. By the end of 2017, these long-term and contingent emission-reduction measures worked jointly to reduce the annually mean $PM_{2.5}$ concentration in Beijing from 89.5 μg/m$^3$ in 2013 to 58.0 μg/m$^3$ in 2017, indicating a great success of $PM_{2.5}$ management during the past five years. The notable decrease of $PM_{2.5}$ concentrations attracted nationwide attentions and growing studies have been conducted to understand spatio-temporal characteristics (Shao et al., 2018; Sun et al., 2019; Wang et al., 2019), sources (Chen et al., 2019; Xu et al., 2019; Cheng, J. et al., 2019) and health effects (Liang et al., 2019) of $PM_{2.5}$ variations in Beijing from 2013 to 2017. These studies revealed that air quality in Beijing was improved significantly in 2017 in terms of annual mean $PM_{2.5}$ concentrations, polluted days and pollution durations. Furthermore, despite different outputs, both source apportionment during pollution episodes based on collected samples (Shao et al., 2019; Xu et al., 2019; Chen et al., 2019) and long-term model simulation based on regional and local emission inventories (Cheng, J. et al., 2019) suggested that local and regional anthropogenic emissions (e.g. coal combustion and vehicle emissions) were the major influencing factors for long-term and short-term $PM_{2.5}$ variations in Beijing.

In addition to anthropogenic emissions, the strong meteorological influences on PM$_{2.5}$ concentrations in Beijing have been widely acknowledged (Zhao et al., 2013; Wang et al., 2014; UNEP, 2016; Cheng et al., 2017; Chen et al., 2017; Sun et al., 2019). For instance, for 2014, more than 180 days in Beijing experienced a dramatic daily AQI (Air Quality Index) change ($\triangle$AQI>50) (Chen, Z. et al., 2016). Considering that anthropogenic emissions for a mega city unlikely changed significantly on a daily basis, rapid variations of meteorological conditions were one major driver for the dramatic change of daily air quality in Beijing. In winter 2017, strong northwest winds led to favorable meteorological conditions for PM$_{2.5}$ diffusion and low PM$_{2.5}$ concentrations in Beijing. This raised the controversy that meteorological conditions, instead of emission-reduction,accounted for the remarkable PM$_{2.5}$ reduction in Beijing. In this case, with the completion of the five-year plan, it is highly necessary to quantify the relative contribution of meteorological conditions and emission-reduction to the notable decrease in PM$_{2.5}$ concentrations in Beijing from 2013 to 2017.

In recent years, growing studies have been conducted to investigate meteorological and anthropogenic influences on long-term PM$_{2.5}$ variations. Based on Goddard Earth Observing System (GEOS) chemical transport model (GEOS-Chem), Yang et al (2016) revealed that the relative contribution of meteorological conditions to PM$_{2.5}$ variations in Eastern China from 1985 to 2005 was 12%. Based on a multiple general linear model (GLM), Gui et al. (2019) quantified that meteorological conditions accounted for 48% of PM$_{2.5}$ variations in Eastern China from 1998 to 2016. Based on a stepwise multiple linear regression (MLR) model, Zhai et al. (2019) quantified the relative contribution of meteorology to PM$_{2.5}$ variations from 2013 to 2018 in Beijing-Tianjin-Hebei region, Yangtze River Delta, Pearl River Delta and Sichuan Basin and Fenwei plain was 14%, 3%, 19%, 27% and 23% respectively. Through a two-stage hierarchical clustering method, Zhang et al. (2018) calculated that the relative contribution of meteorological conditions to heavy pollution episodes within the Beijing-Tianjin-Hebei region was larger than 50% from 2013 to 2017. These studies quantified the overall meteorological influences on long-term PM$_{2.5}$ variations using different statistical models and chemical transport models (CTMs). However, due to strong interactions between individual meteorological factors, traditional statistical methods such as correlation analysis and linear regression may be biased significantly when quantifying meteorological influences on PM$_{2.5}$ concentrations (Chen et al., 2017). On the

other hand, the accuracy of CTMs can be influenced largely by the uncertainty in emission
inventories (Xu et al., 2016) and deficiency of heterogeneous/aqueous processes (Li et al.,
2011). Therefore, multiple advanced models should be comprehensively considered to better
quantify meteorological influences on $PM_{2.5}$ concentrations (Pearce et al., 2011).
To evaluate this five-year clean-air plan, we employ an advanced statistical model,
Kolmogorov-Zurbenko (KZ) filtering, which is advantageous of filtering meteorological
influences on long-term time series of airborne pollutants, and a CTM model, WRF-CMAQ,
which is advantageous of quantifying the relative contribution of different emission sources,
to comprehensively investigate the relative contribution of meteorological conditions and
emission-reduction to $PM_{2.5}$ reduction in Beijing from 2013 to 2017 respectively. In this light,
this research provides important insight for better designing and implementing successive
clean air plans in the future to further mitigate $PM_{2.5}$ pollution in Beijing.
This manuscript is structured as follows: Firstly, major data sources, including $PM_{2.5}$ and
meteorological data, and emission inventories, employed for this research are briefly
introduced. Secondly, the principle and parameter setting of two models, KZ filtering and
WRF-CMAQ, and model verification are explained. In the result section, the relative
contribution of meteorological conditions and anthropogenic emissions to $PM_{2.5}$ variations in
Beijing from 2013 to 2017 calculated using both models is presented. In the discussion and
conclusion part, implementations of this research and suggestions for further improving air
quality in Beijing are given.
**2 Data Sources**
**2.1 $PM_{2.5}$ and meteorological data**
In this study, hourly $PM_{2.5}$ concentration data were acquired from the website PM25.in
([www.PM25.in](www.PM25.in)), which collects official data provided by China National Environmental
Monitoring Center (CNEMC). Beijing has established an advanced air quality monitoring
network with 35 ground stations across the city. Considering the major contribution of
industry and traffic-induced emissions in urban areas, we selected all twelve urban stations
to analyze spatio-temporal variations of $PM_{2.5}$ concentrations and quantify their influencing
factors. In addition to these urban stations, we selected two background stations, the
DingLing Station located in the suburb and the MiYun Reservoir Station located in the outer
suburb, one transportation station (the Qianmen station) located close to a main road, and
one rural station (the Yufa Station) that is far away from central Beijing for the following
analysis. The DingLing and MiYun Reservoir Station were chosen as background stations by
the Ministry of Environmental Protection of China. These two stations receive limited
influence from anthropogenic emissions due to their location in suburban and outer suburban
areas. The Qianmen transportation station received more influences from vehicle emissions.
Long-term variations of $PM_{2.5}$ concentrations in different type of stations provide a useful
reference for comprehensively understanding the effects of emission-reduction measures on
$PM_{2.5}$ decrease in Beijing from 2013 to 2017. Meteorological data for this research were
collected from the Guanxiangtai Station (GXT,54511, 116.46° E, 39.80° N), Beijing and
downloaded from the Department of Atmospheric Science, College of Engineering,
University of Wyoming (http://weather.uwyo.edu/upperair/sounding.html). Both $PM_{2.5}$ and
meteorological data were collected from January 1st, 2013 to December 31st, 2017. The
locations of these selected stations are shown in Fig 1.

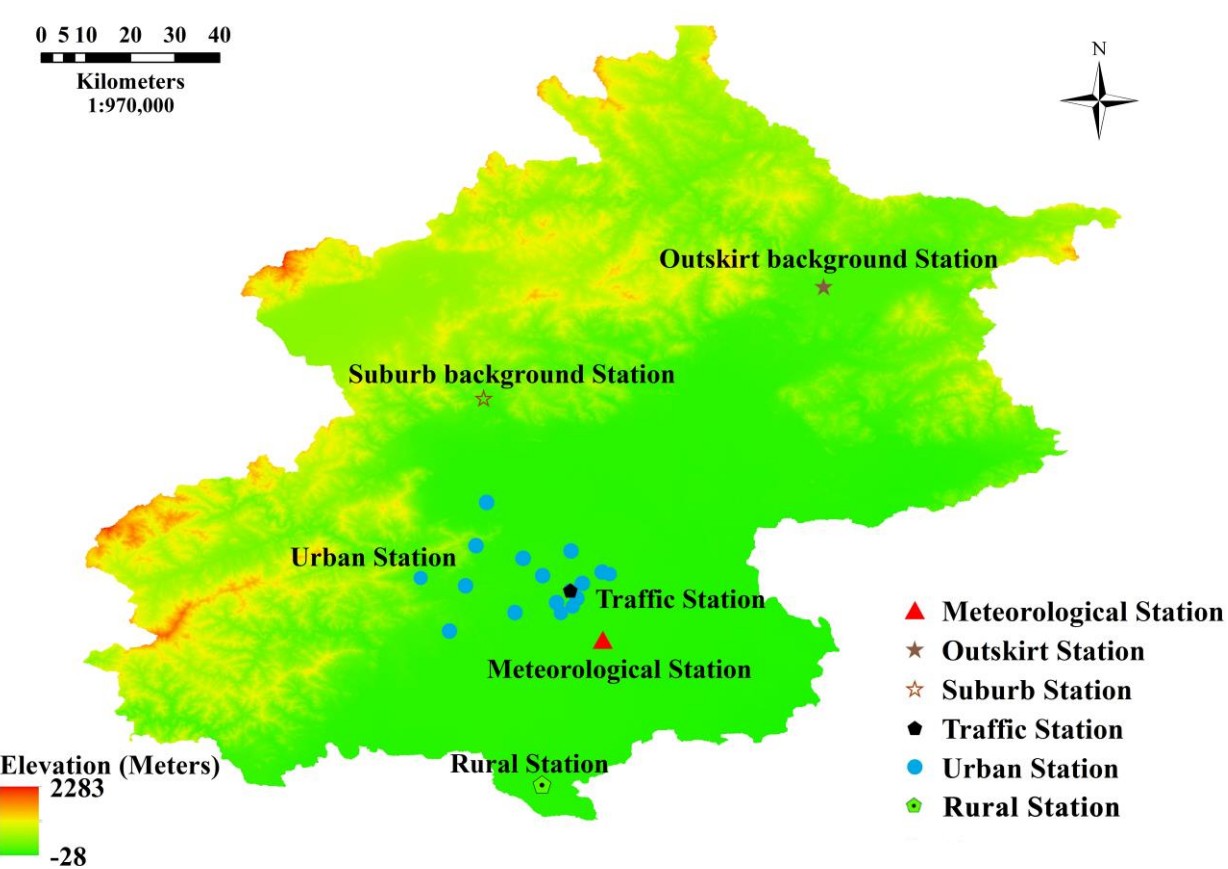


**Fig 1. Locations of different ground monitoring stations.**

**2.2 Emission inventories**

For this research, we employed both regional and local emission inventories for running model simulation. Multi-resolution Emission Inventory for China, MEIC, (http://meicmodel.org/) provided by Tsinghua University, were employed as the regional emission inventories. MEIC has been widely employed and verified as a reliable emission inventory by a diversity of studies (Hong et al., 2017; Saikawa et al., 2017; Zhou et al., 2017; etc.). For simulating five-year $PM_{2.5}$ concentrations, MEIC from 2013 to 2017 are required. Since official MEIC 2017 has not been available yet, we employed a strategy from previous studies (Chen et al., 2019; etc) and updated MEIC 2016 for simulating emission-reduction scenarios and $PM_{2.5}$ concentrations in 2017 by considering official 2017 emission-reduction plans (e.g. the target of coal combustion reduction) required by the local government.

Different from regional emission inventories, local emission inventories are usually produced independently by local institutions. The Beijing local-emission inventory employed for this research was produced and updated by Beijing Municipal Research Institute of Environmental protection, fully according to the requirement of MEP on the production of local emission inventories within Beijing-Tianjin-Hebei region. This Beijing local-emission inventory from 2013 to 2017 was produced by synthesizing local environmental statistical data and reported emission data, carrying out field investigations and conducting a series of estimation according to Beijing Five-year Clean Air Action Plan. As shown in table 1, it is highly consistent with other official statistical data, such as the Annual report from National Environmental Statistics Bulletin (**http://www.mee.gov.cn/gzfw_13107/hjtj/qghjtjgb/**) and "2+26" Center for Air Pollution Prevention and Control, and has been formally employed for the implementation of recent "2017 Air Pollution Prevention and Management Plan for the Beijing-Tianjin-Hebei Region and its Surrounding Areas" (MEP, 2017).

**Table 1. The comparison of local environmental statistical data used for this research**

**and other official statistical data in 2017 (unit: 10k tons)**

| | SO$_2$ | NO$_x$ | CO | VOC | NH$_3$ | PM$_{10}$ | PM$_{2.5}$ | BC | OC |
|---|---|---|---|---|---|---|---|---|---|
| **Statistical data for this research** | 1.38 | 10.15 | 49.54 | 13.47 | 3.20 | 14.74 | 3.92 | 0.17 | 0.44 |
| **National Environmental Statistics Bulletin** | 1.38 | 12.16 | 52.03 | 24.24 | 3.26 | 14.68 | 3.91 | 0.22 | 0.41 |
| **"2+26" center for air pollution prevention and control** | 0.89 | 9.24 | 48.98 | 13.93 | 3.16 | 13.82 | 3.72 | 0.19 | 0.46 |

## 3 Methods

A key step for quantifying the relative contribution of anthropogenic emissions to PM$_{2.5}$ variations is to properly filter meteorological influences on PM$_{2.5}$ concentrations, which is highly challenging and rarely investigated by previous studies. Therefore, we employed both a statistical method and a CTM to comprehensively evaluate the role of anthropogenic emissions and meteorological conditions in the decrease of PM$_{2.5}$ concentrations in Beijing from 2013 to 2017.

### 3.1 Kolmogorov-Zurbenko (KZ) filtering

Since meteorological conditions exert a strong influence on PM$_{2.5}$ concentrations in Beijing, the removal of seasonal signals from time series of meteorological factors produces data sets suitable for understanding the trend of PM$_{2.5}$ concentrations mainly influenced by anthropogenic factors (Eskridge et al., 1997). To better analyze the trend of time series data without the disturbances from other major influencing variables, a statistical method Kolmogorov-Zurbenko (KZ) filtering was proposed by Rao et al. (1994). The KZ filter is advantageous of removing high-frequency variations in data sets through iterative moving average. Eskridge et al. (1997) compared four major approaches for trend detection, including PEST, anomalies, wavelet transform, and the KZ filter, and suggested that KZ achieved higher confidence in detecting long-term trend than other models. Due to its reliable performance in trend detection in complicated ecosystems, the KZ filter has been increasingly employed to remove seasonal signals of meteorological conditions and extract long-term trend of airborne pollutants (Zurbenko, et al., 1996; Eskridge, et al., 1997; Kang,

et al., 2013; Ma et al., 2016; Cheng, N et al., 2019). One potential limitation of the KZ filter
is that iterative moving average ($m$) may impose an influence on detecting abrupt variations.
Therefore, Zurbenko et al.(1996)proposed an enhanced KZ filter that employed a dynamic
variable $m$ that decreased with the increase in changing rate. For this research, we employed
this dynamic m to produce an adjusted time-series of PM$_{2.5}$ concentrations in Beijing by
removing large inter-annual and seasonal variations in meteorological conditions. The
principle of the KZ filter is briefly introduced as follows.
The raw time-series of airborne pollutants can be decomposed as:

$$X(t)=E(t)+S(t)+W(t) \qquad (1)$$

$$X_b(t) = E(t) + S(t) \qquad (2)$$

$$E(t) = KZ_{365,3}(X) \qquad (3)$$

$$S(t) = KZ_{15,5}(X) - KZ_{365,3}(X) \qquad (4)$$

$$W(t) = X(t) - KZ_{15,5}(X) \qquad (5)$$

**Where X ($t$) is the original time series of airborne pollutants, E($t$) is the long-term trend component,**
**S($t$) is the seasonal component, W($t$) is the short-term (synoptic-scale) component or residue. KZ$_{i, j}$(X)**
**indicates KZ filtering on the original dataset X with a moving wind size of $i$ and $j$ iterations.**
$X_b$ ($t$) stands for the base component, the sum of the long-term and seasonal component,
presenting steady trend variation. E($t$) is mainly affected by long-term anthropogenic
emission and climate change. S($t$) is mainly influenced by the seasonal variation of emission
and meteorological conditions. W($t$) is caused by short-term and small-scale shifts of
emissions and meteorological conditions.
The long-term trend component E($t$) processed by KZ filtering still contains the influence of
meteorological conditions, which can be removed by multiple regression models. Multiple
linear relationships are established for the residue and baseline component respectively using
meteorological factors strongly correlated with airborne pollutants.
We examined correlations between seasonal PM$_{2.5}$ concentrations in Beijing and a series of
meteorological factors, including temperature, wind speed, wind direction, precipitation,
relative humidity, solar radiation, evaporation and air pressure. Due to limited space, detailed
correlations between PM$_{2.5}$ concentrations and individual meteorological factors in Beijing
are not presented here and readers can refer to previous studies for more information (Chen
et al., 2017; 2018). The correlation analysis revealed that wind speed, relative humidity,
temperature and solar radiation were strongly and significantly correlated with $PM_{2.5}$
concentrations in Beijing (as shown in Table 2), which was consistent with findings from
other studies (Sun et al., 2013; Wang et al., 2018).
**Table 2. Major meteorological factors strongly correlated with seasonal $PM_{2.5}$**
**concentrations in Beijing (Chen et al., 2017)**

| Spring | Summer | Autumn | Winter |
|---|---|---|---|
| | RHU**(0.648) | RHU**(0.587) | RHU**(0.738) |
| RHU**(0.532) | SSD**(−0.447) | SSD**(−0.509) | SSD**(−0.715) |
| | TEM**(0.554) | WIN**(−0.468) | WIN**(−0.558) |

**\*\*Correlation is significant at the 0.01 level (2 tailed);**
**RHU: Relative humidity; SSD: Sunshine Duration; TEM: Temperature; WIN: Wind speed**
Therefore, we further established multiple linear regression equations between $PM_{2.5}$
concentrations and wind speed, relative humidity, temperature and solar radiation as follows.
$$W(t) = \alpha_0 + \sum \alpha_i \, w_i(t) + \varepsilon_w(t) \quad (6)$$

$$X_b(t) = b_0 + \sum b_i \, x_i(t) + \varepsilon_b(t) \quad (7)$$

$$\varepsilon(t) = \varepsilon_w(t) + \varepsilon_b(t) \quad (8)$$

**Where $w_i(t)$ and $x_i(t)$ stand for the different short-term and baseline component of the $i^{th}$**
**meteorological factor. $\varepsilon_w$ and $\varepsilon_b$ is the regression residue of the short-term and baseline**
**component. $\varepsilon(t)$ indicates the total residue, including the short-term influence of local emission**
**and meteorological factors neglected during the regression process and other noises.**
Next, KZ filtering was conducted on the $\varepsilon(t)$ for its long-term component $\varepsilon_E(t)$. After the
variation of meteorological influences was filtered, the reconstructed time series of airborne
pollutants $X_{LT}(t)$ was calculated as the sum of $\varepsilon_E(t)$ and the average value of E(t), $\overline{E(t)}$.
$$X_{LT}(t) = \overline{E(t)} + \varepsilon_E(t) \quad (9)$$

After KZ filtering, the relative contribution of meteorological conditions to $PM_{2.5}$ variations
can be calculated as follows:
$$P_{contrib} = \frac{K_{org} - K}{K_{org}} \times 100\% \quad (10)$$

**Where $P_{contrib}$ is the relative contribution of meteorological conditions to PM$_{2.5}$ variations in Beijing,**
**$K_{org}$ is the variation slope of the original PM$_{2.5}$ time series; K is the variation slope of adjusted PM$_{2.5}$**
**time series with filtered influences from meteorological variations.**
**3.2 WRF-CMAQ model**
We employed WRF-CMAQ for simulating the effects of emission-reduction on the decrease
of PM$_{2.5}$ concentrations. WRF-CMAQ includes three models: The middle-scale meteorology
model (WRF), the source emission model (SMOKE) (http://www.cmascenter.org/smoke/)
and the community multiscale air quality modeling system (CMAQ)
(http://www.cmascenter.org/CMAQ). The center of the CMAQ was set at coordinate 35°N,
110°E and a bi-directional nested technology was employed, producing two layers of grids
with a horizontal resolution of 36 km and 12 km respectively. The first layer of grids with
36km resolution and 200×160 cells covered most areas in East Asia (including China, Japan,
North Korea, South Korea, and other countries). The second layer of grids with 12km
resolution and 120×102 cells covered the North China Plain (including the
Beijing-Tianjin-Hebei region, Shandong and Henan Province). The vertical layer was
divided into 20 unequal layers, eight of which were of a less-than-1km distance to the
ground for better featuring the structure of atmospheric boundary. The height of the ground
layer was 35m.
We employed ARW-WRF3.2 to simulate the meteorological field. The setting of the center
and the bidirectional nest for WRF and CMAQ was similar. There were 35 vertical layers for
WRF and the outer layer provided boundary conditions of the inner layer. The
meteorological background field and boundary information with a FNL resolution of 1°×1°
and temporal resolution of 6h were acquired from NCAR (National Center for Atmospheric
Research, https://ncar.ucar.edu/) and NCEP (National Centers for Environmental Prediction)
respectively. The terrain and underlying surface information was obtained from the USGS
30s global DEM (https://earthquake.usgs.gov/). The outputs from WRF were interpolated to
the region and grid of CMAQ using the Meteorology-Chemistry Interface Processor (MCIP,
https://www.cmascenter.org/mcip). The meteorological factors used for this model included
temperature, air pressure, humidity, geopotential height, zonal wind, meridional wind,
precipitation, boundary layer heights and so forth. An estimation model for terrestrial
ecosystem MEGAN (http://ab.inf.uni-tuebingen.de/software/megan/) was employed to
process the natural emissions. Multi-resolution Emission Inventory for China, MEIC
0.5°×0.5° emission inventory (http://www.meicmodel.org/) and Beijing emission inventory
(http://www.cee.cn/) provided anthropogenic emission data. We input the processed natural
and anthropogenic emission data into the SMOKE model and acquired comprehensive
emission source files.
Scenario simulation is employed to estimate the contribution of emission-reduction to the
variation of PM$_{2.5}$ concentrations.

$$P_{contrib} = \frac{C - C_{base}}{C} \times 100\% \quad (11)$$

**Where $P_{contrib}$, $C$ and $C_{base}$ are the contribution rate of emission-reduction to PM$_{2.5}$**
**concentrations, simulated PM$_{2.5}$ concentrations under the emission-reduction scenario, and**
**simulated PM$_{2.5}$ concentrations in the baseline scenario respectively.**
To evaluate the relative contribution of meteorological conditions and different
emission-reduction measures to the decrease of PM$_{2.5}$ concentrations, we designed two
baseline experiments and four sensitivity experiments. For the first baseline experiment, we
employed the actual meteorological data in 2013. For the second baseline experiment, we
employed the actual meteorological data in 2017 and emission inventory in 2017. Since no
emission-reduction measures were conducted in 2013, the first baseline experiment was used
to estimate the relative contribution of meteorological conditions to the variation of PM$_{2.5}$
concentrations. By comparing the first and second baseline experiment, the relative
contribution of all emission-reduction measures to the variation of PM$_{2.5}$ concentrations can
be quantified. For the first sensitivity experiment, we employed the actual meteorological
conditions in 2013 and emission inventory in 2017 and compared the simulation result with
the baseline experiment, which demonstrated the relative contribution of meteorological
concentrations to PM$_{2.5}$ reduction in Beijing from 2013 to 2017. Since the WRF-CMAQ
simulation simply considers PM$_{2.5}$ concentrations and meteorological conditions in 2013 and
2017 without considering their variation process from 2013 to 2017, KZ filtering may
perform better in quantifying the relative contribution of meteorological variations to PM$_{2.5}$
reduction in Beijing. However, the output from this sensitivity experiment serves as a useful
reference for cross-verifying the output from the KZ filtering. For the remaining three
sensitivity-simulation experiments, we added the reduced emission amount induced by one
specific emission-reduction measure to the actual emission amount in 2017 and kept other
parameters unchanged, and thus quantified the relative contribution of one specific
emission-reduction measure to $PM_{2.5}$ reduction in Beijing from 2013 to 2017. Consequently,
we quantified the relative contribution of three major emission-reduction measures to $PM_{2.5}$
reduction in Beijing (Table 3).
**Table 3.   The design and materials for two baseline and four sensitivity experiments using WRF-CMAQ**

| ID | Meteorological Data | Emission-reduction measures | Simulation Year | Major purposes |
|---|---|---|---|---|
| **Baseline Experiment1** | 2013 | No emission-reduction Measures | 2013 | **2013 baseline scenario** |
| **Baseline Experiment2** | 2017 | All emission-reduction Measures | 2017 | **2017 baseline scenario** |
| **Sensitivity Experiment 1** | 2013 | All emission-reduction Measures | 2017 | The relative contribution of meteorological variations to the decrease of $PM_{2.5}$ concentrations in Beijing from 2013 to 2017 |
| **Sensitivity Experiment 2** | 2017 | All emission-reduction measures except for industrial restructuring | 2017 | The relative contribution of industrial restructuring to the decrease of $PM_{2.5}$ concentrations in Beijing from 2013 to 2017 |
| **Sensitivity Experiment 3** | 2017 | All emission-reduction measures except for the regulation of coal boilers | 2017 | The relative contribution of the regulation of coal boilers to the decrease of $PM_{2.5}$ concentrations in Beijing from 2013 to 2017 |
| **Sensitivity Experiment 4** | 2017 | All emission-reduction measures except for increasing clean fuels for civil use | 2017 | The relative contribution of increasing clean fuels for civil use to the decrease of $PM_{2.5}$ concentrations in Beijing from 2013 to 2017 |

**For emission data, all experiments employed Beijing local emissions inventory in 2017 for Beijing and regional emission inventory in 2017 for other regions.**
**MEIC 2017 was acquired based on our update of MEIC 2016 according to official 2017 emission-reduction targets required by the local government.**

**3.3 Model verification**

**3.3.1 Verification of KZ filtering**

For each station, the original time series of $PM_{2.5}$ data was processed by the KZ filter and the relative contribution of the long-term, seasonal and short-term component to the total variance is shown as Table 4. The sum of the long-term, seasonal and short-term component contributed to more than 93.6~95.3% of the total variance in different stations respectively. The larger the total variance, the three components are more independent to each other. The total variance close to 100% suggests that a majority of meteorological influences has been considered and effectively removed. As shown in Table 4, the large value of the total variation in all stations indicated a satisfactory output from the KZ filtering.

Specifically, the relative contribution of the seasonal component (ranging from 9%-23.8%) and short-term component (ranging from 66.8%-83.8%) was much larger than that of the long-term component (ranging from 1.2%-3.5%), suggesting that seasonal and short-term variations of meteorological and emission factors exerted a major influence on the rapid change of $PM_{2.5}$ concentrations in Beijing. The decomposed long-term, seasonal and short-term component from the original time series of mean urban $PM_{2.5}$ concentrations in Beijing from 2013 to 2017 are demonstrated as Fig 2. According to Fig 2, the notable peaks of decomposed seasonal and short-term component were highly consistent with the peaks of $PM_{2.5}$ concentrations in the original time-series, which further proved the dominant influence of seasonal and short-term variations of meteorological and anthropogenic factors on the temporal changes of $PM_{2.5}$ concentrations in Beijing.

**Table 4. The relative contribution of different components to the total variance of**

**original time series of PM$_{2.5}$ concentrations from 2013-2017 at different stations**

| Stations | Long-term component (%) | Seasonal component (%) | Short-term component (%) | Total variance(%) |
|---|---|---|---|---|
| **Yufa** | 2.1 | 23.8 | 66.8 | 94.0 |
| **Miyun Reservoir** | 1.4 | 9.0 | 83.8 | 95.2 |
| **Dingling** | 1.6 | 11.0 | 81.3 | 94.9 |
| **Qianmen** | 2.7 | 12.7 | 78.5 | 95.1 |
| **Olympic center** | 2.1 | 11.9 | 80.0 | 95.3 |
| **Xiangshan** | 1.2 | 10.3 | 83.4 | 94.9 |
| **Huayuan** | 2.2 | 15.9 | 75.6 | 93.7 |
| **Yungang** | 2.1 | 15.1 | 76.5 | 93.6 |
| **WanShouxigong** | 1.6 | 14.2 | 78.2 | 94.0 |
| **Dongsi** | 1.6 | 12.3 | 80.0 | 94.0 |
| **TianTan** | 2.1 | 13.2 | 78.6 | 93.8 |
| **NongZhanguan** | 1.8 | 13.7 | 78.6 | 94.1 |
| **Gucheng** | 1.8 | 13.5 | 78.5 | 93.7 |
| **Guanyuan** | 1.6 | 12.6 | 79.8 | 94.0 |
| **BeiBuxinqu** | 1.7 | 13.8 | 78.4 | 93.9 |
| **WanLiu** | 3.5 | 11.9 | 78.2 | 93.6 |

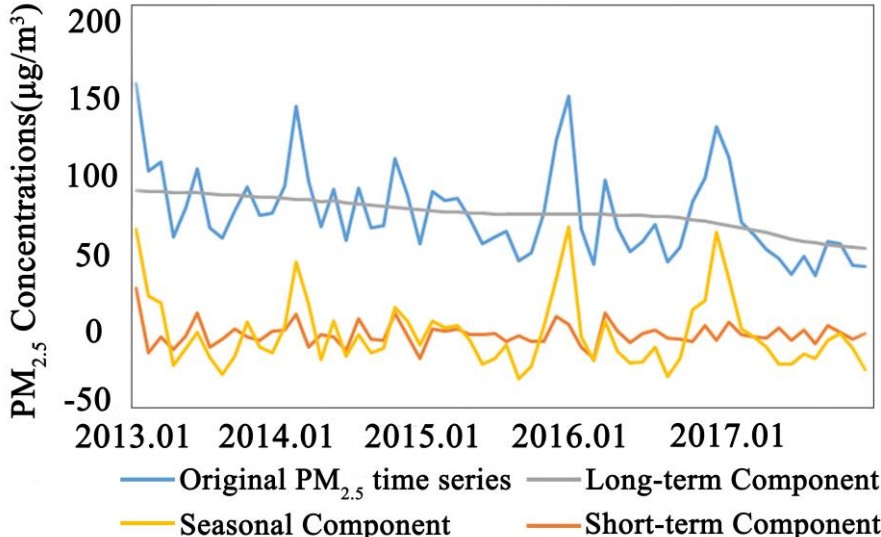

**Fig 2. The long-term, seasonal and short-term component extracted from the original time series of mean urban PM$_{2.5}$ concentrations in Beijing from 2013 to 2017**

**3.3.2 Verification of WRF-CMAQ**

We employed the emission inventory and meteorological data for 2017 to verify the accuracy of WRF-CMAQ simulation. For six stations of different types (DingLing background station, Yufa rural station, Olympic Center urban station, Guanyuan urban station, Dongsi urban station and Agricultural museum urban station), we compared the observed and estimated PM$_{2.5}$ concentrations and presented the comparison result as Fig 3. According to Fig 3, the general trend of the simulated PM$_{2.5}$ concentrations was consistent with that of the observed PM$_{2.5}$ concentrations. For six stations, the correlation coefficient R, normalized mean bias (NMB), normalized mean error (NME), mean fractional bias (MFB) and mean fractional error (MFE) between observed and simulated data was 0.63~0.91, -6%~6%, 26%~40%, -5%~7%, and 27%~46% respectively, indicating a satisfactory simulation output (EPA,2005; Boylan et al., 2006). However,as shown in Figure 3, WRF-CMAQ may notably underestimate PM$_{2.5}$ concentrations during heavy pollution episodes due to unified parameter setting for long-term simulation, the uncertainty in emission inventories, and especially insufficient chemical reaction mechanisms, which is a common challenge for CTM-based PM$_{2.5}$ simulation (Li et al., 2011). For instance, without considering heterogeneous/aqueous reactions between multiple precursors, CTMs failed to approach the maximum PM$_{2.5}$ concentrations during severe haze episodes

and the simulation accuracy was dramatically improved by including proper
descriptions of heterogeneous/aqueous reactions into CTMs (Chen, D. et al. 2016).
With more finer-scale emission inventories and better descriptions of reaction
mechanisms between precursors, the accuracy of PM$_{2.5}$ simulation can be improved
significantly.

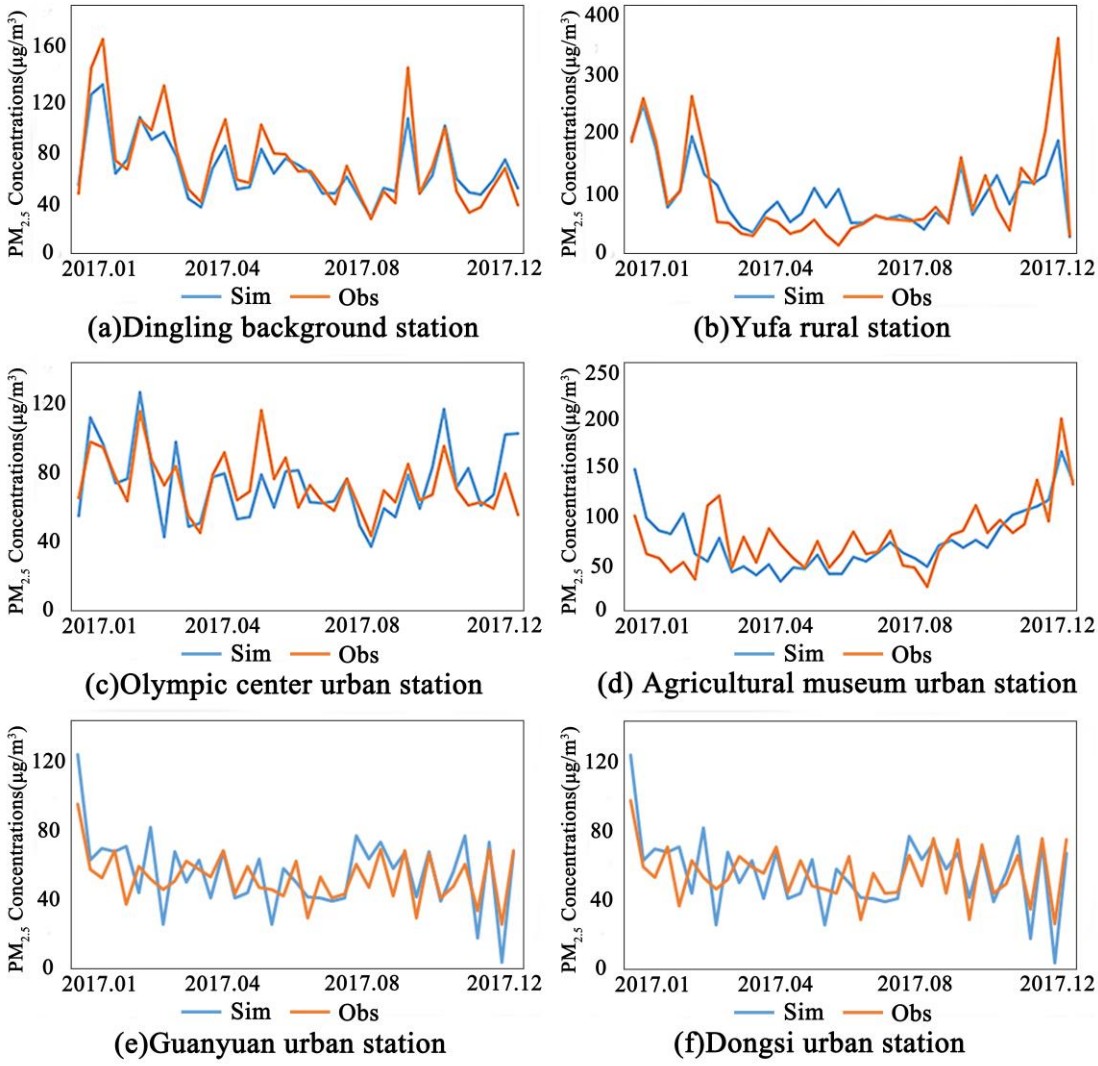


**Fig 3. The comparison between observed and WRF-CMAQ simulated PM$_{2.5}$**
**concentrations in 2017 in six stations across Beijing**

## 4 Results

### 4.1 The relative contribution of emission-reduction and meteorological variations to the decrease of PM$_{2.5}$ concentrations in Beijing from 2013 to 2017

#### 4.1.1 Estimation based on KZ filtering

Through KZ filtering, the adjusted time-series of PM$_{2.5}$ concentrations with filtered meteorological variations was acquired. Next, for each station, the actual PM$_{2.5}$ variations and adjusted PM$_{2.5}$ variations without the disturbance of meteorological variations from 2013 to 2017 were calculated respectively (as shown in Table 5). Based on this, the relative contribution of emission-reduction and meteorological conditions to PM$_{2.5}$ reduction in Beijing from 2013 to 2017 can be quantified.

The original and KZ-processed time series of PM$_{2.5}$ concentrations were illustrated using one background station, one rural station and four urban stations (Fig 4). As shown in Fig 4, most abrupt variations in the original time series of PM$_{2.5}$ concentrations have been smoothed through KZ filtering and the generally decreasing trend of PM$_{2.5}$ variations from 2013 to 2017 caused by anthropogenic emissions can be clearly presented.

       **Table 5. Estimated relative contribution of emission-reduction and meteorological variations to PM$_{2.5}$ reduction in Beijing from 2013 to 2017 using KZ filter**

| Stations | PM$_{2.5}$ concentrations in 2013(µg·m$^{-3}$) | PM$_{2.5}$ concentrations in 2017 (µg·m$^{-3}$) | Adjusted PM$_{2.5}$ concentrations in 2017(µg·m$^{-3}$) | PM$_{2.5}$ Decrease rate (µg·m$^{-3}$·m$^{-1}$)[1] | Adjusted PM$_{2.5}$ Decrease rate (µg·m$^{-3}$·m$^{-1}$)[2] | Contribution of emission reduction (%)[3] | Contribution of meteorological variations (%)[4] |
|---|---|---|---|---|---|---|---|
| Yufa | 111.1 | 69.7 | 74.6 | -0.78 | -0.63 | 80.4 | 19.7 |
| Miyun Reservoir | 58.8 | 44.8 | 47.0 | -0.40 | -0.33 | 82.8 | 17.2 |
| Dingling | 69.6 | 47.1 | 50.6 | -0.54 | -0.44 | 80.8 | 19.2 |
| Qianmen | 103.9 | 64.0 | 68.9 | -0.81 | -0.69 | 85.0 | 15.0 |
| Olympic center | 90.4 | 57.2 | 61.7 | -0.68 | -0.55 | 80.8 | 19.2 |
| Xiangshan | 77.0 | 59.3 | 60.3 | -0.46 | -0.39 | 83.9 | 16.1 |
| Huayuan | 101.5 | 64.4 | 69.2 | -0.77 | -0.63 | 81.9 | 18.1 |
| Yungang | 91.8 | 60.2 | 64.0 | -0.69 | -0.55 | 79.6 | 20.4 |
| WanShouxigong | 93.7 | 62.0 | 66.8 | -0.64 | -0.50 | 78.2 | 21.8 |
| Dongsi | 94.9 | 62.4 | 67.5 | -0.62 | -0.49 | 78.9 | 21.1 |
| TianTan | 92.3 | 58.4 | 64.6 | -0.68 | -0.55 | 80.2 | 19.9 |
| NongZhanguan | 92.2 | 59.9 | 65.9 | -0.66 | -0.53 | 80.3 | 19.8 |
| Gucheng | 92.7 | 61.4 | 65.9 | -0.65 | -0.50 | 77.6 | 22.4 |
| Guanyuan | 89.6 | 59.5 | 64.6 | -0.60 | -0.48 | 79.6 | 20.4 |
| BeiBuxinqu | 86.6 | 59.5 | 63.3 | -0.60 | -0.45 | 75.2 | 24.8 |
| WanLiu | 98.1 | 56.2 | 60.4 | -0.87 | -0.73 | 84.2 | 15.8 |

   [1] PM$_{2.5}$ decrease rate: the fitted variation slope of original monthly average PM$_{2.5}$ time series;

   [2] Adjusted PM$_{2.5}$ decrease rate: the fitted variation slope of adjusted monthly average PM$_{2.5}$ time series;

   [3.] Contribution of emission reduction = 1 - Contribution of meteorological variations;

   [4.] Contribution of meteorological variations = (PM$_{2.5}$ decrease rate - Adjusted PM$_{2.5}$ decrease rate) / PM$_{2.5}$ decrease rate.

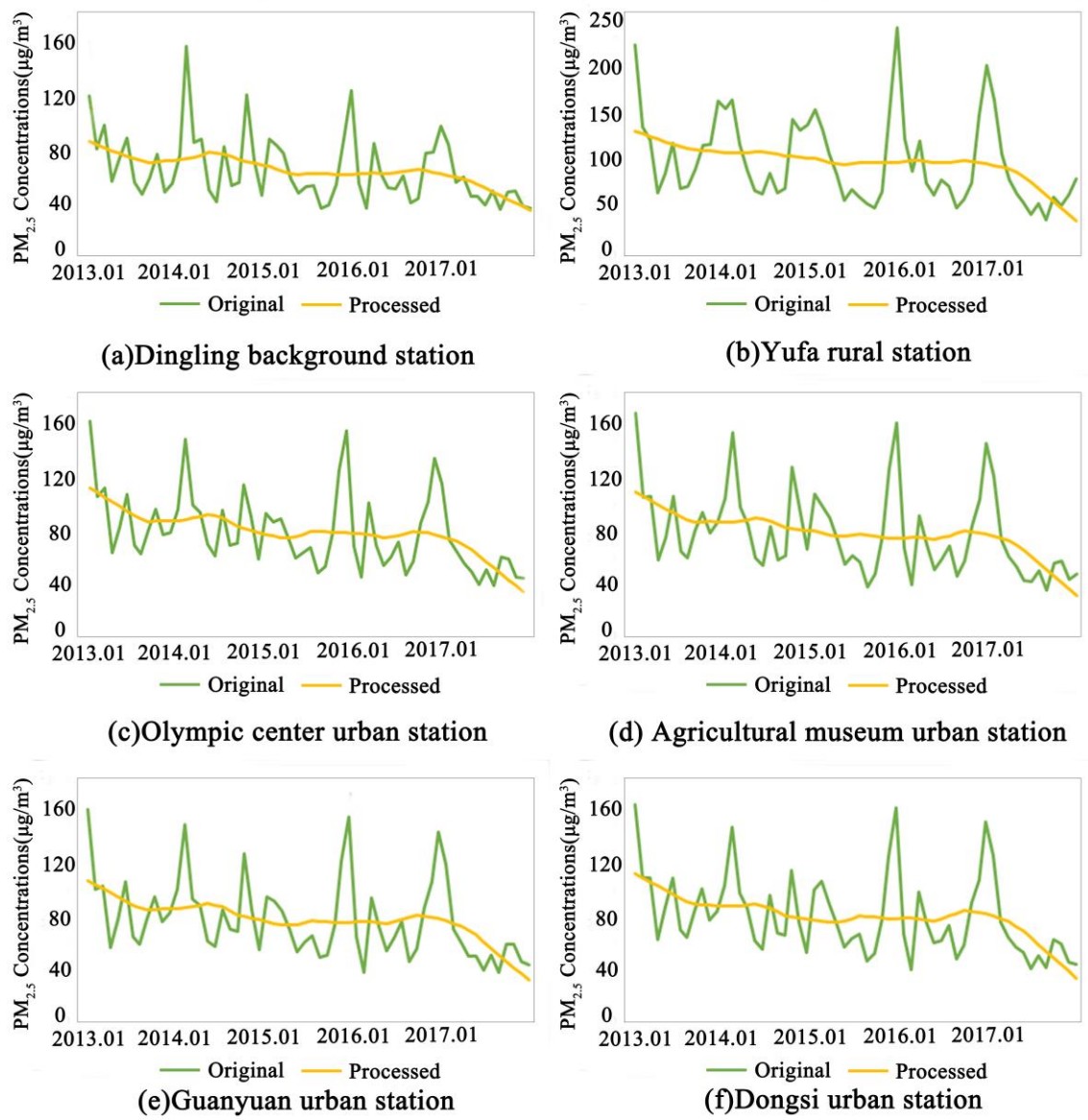

**Fig 4. The comparison of original and KZ processed time series of PM$_{2.5}$ concentrations in six stations from 2013 to 2017**

According to Table 5, the annual mean PM$_{2.5}$ concentration in Beijing in 2017 was 35.6% lower than that in 2013. By filtering the influence of meteorological variations, the adjusted annual mean PM$_{2.5}$ concentration in Beijing in 2017 decreased by 31.7% when compared to that in 2013, indicating that the variation in meteorological conditions exerted a moderate influence on PM$_{2.5}$ reduction from 2013 to 2017. Meteorological conditions in Beijing were generally favorable for PM$_{2.5}$ dispersion during the five-year period, especially the latter half of 2017, when there was a high frequency of strong northerly winds and much lower wintertime PM$_{2.5}$ concentrations than previous years.

For the winter of 2017, frequent windy weather and successive clean sky had a strong
influence on the reduction of $PM_{2.5}$ concentrations in Beijing. This led to a hot debate
concerning whether the notable decrease in $PM_{2.5}$ concentrations was mainly
attributed to the favorable meteorological conditions or emission-reduction. Table 5
suggests that the control of anthropogenic emissions contributed to 75.2%~85.0% of
$PM_{2.5}$ decrease in the five-year period, indicating that emission-reduction worked
effectively in all rural, urban and background stations. On average, the relative
contribution of emission-reduction and meteorological variations to $PM_{2.5}$ reduction
in Beijing from 2013 to 2017 was 80.6% and 19.4% respectively. Therefore, in spite
of more favorable meteorological conditions, properly designed and implemented
emission-reduction measures were the dominant driver for the remarkable decrease of
$PM_{2.5}$ concentrations in Beijing from 2013 to 2017.
**4.1.2 Estimation based on WRF-CMAQ**
In addition to the KZ filter, we also employed WRF-CMAQ to estimate the relative
contribution of emission-reduction and meteorological conditions to the decrease of
$PM_{2.5}$ concentrations in Beijing. The result is shown in Table 6.
**Table 6. Estimated relative contribution of emission-reduction and meteorological variations to**
**$PM_{2.5}$ reduction in Beijing from 2013 to 2017 using WRF-CMAQ**

| Stations | Contribution of meteorological variations (%) | Contribution of emission-reduction(%) |
|---|---|---|
| **Yufa** | 21.9 | 78.2 |
| **Miyun Reservoir** | 20.8 | 79.2 |
| **Dingling** | 21.7 | 78.3 |
| **Qianmen** | 21.2 | 78.8 |
| **Olympic center** | 21.2 | 78.8 |
| **Xiangshan** | 20.3 | 79.7 |
| **Huayuan** | 21.2 | 78.8 |
| **Yungang** | 21.2 | 78.8 |
| **WanShouxigong** | 21.2 | 78.8 |
| **Dongsi** | 21.2 | 78.8 |
| **TianTan** | 21.2 | 78.8 |
| **NongZhanguan** | 21.2 | 78.8 |
| **Gucheng** | 22.2 | 77.8 |
| **Guanyuan** | 21.2 | 78.8 |
| **BeiBuxinqu** | 22.2 | 77.8 |
| **WanLiu** | 22.2 | 77.8 |

Based on WRF-CMAQ, the relative contribution of meteorological variations to the
decrease in $PM_{2.5}$ concentrations in Beijing ranged from 20.3% to 22.2% in different
stations, whilst emission-reduction accounted for about four-fifths of $PM_{2.5}$ reduction
from 2013 to 2017. It is worth mentioning that WRF-CMAQ is a grid-based model
and thus the calculated contribution of meteorological variations for some stations
located in the same grid was the same. Instead, station-based KZ filtering led to more
reliable analysis for each station and can better distinguish the differences between
multiple stations. Furthermore, WRF-CMAQ simply considered the differences
between meteorological conditions in 2013 and 2017 without considering their
variations during the five-year period while the KZ filtering analyzed the entire time
series of $PM_{2.5}$ and meteorological data from 2013 to 2017. The averaged relative
contribution of meteorological variations to $PM_{2.5}$ reduction in Beijing calculated
using WRF-CMAQ was 21.4%, very similar to the 19.4% calculated using KZ
filtering. The slightly larger meteorological contribution calculated using
WRF-CMAQ might be attributed to that WRF-CMAQ simply considered the
favorable meteorological conditions in 2017 whilst KZ fully considered the long-term
meteorological variations from 2013 to 2017.
Since KZ filtering is fully based on observed data, and simply considers the influence
of time-series meteorology data on $PM_{2.5}$ time series, less uncertainty is involved. The
accuracy of KZ filtering is influenced mainly by the variations of $PM_{2.5}$-meteorology
interactions in different areas and seasons. On the other hand, CTMs (e.g.
WRF-CMAQ or WRF-CAMx) consider both meteorological conditions (mainly
large-scale meteorological data for model simulation, not as accurate as local
observed meteorological data) and anthropogenic emissions for estimating $PM_{2.5}$
concentrations under different emission scenarios. The accuracy of these models are
not only decided by proper understanding of $PM_{2.5}$-meteorology interactions, but also
the reliability of emission inventories and proper descriptions of reaction mechanisms
for $PM_{2.5}$ production, especially during heavy pollution episodes, which is a major
challenge for current model simulation. Consequently, KZ filtering provides a more
reliable method for researchers and decision makers to understand the relative
importance of emission-reduction and meteorological conditions in recent $PM_{2.5}$
reduction in Beijing. Meanwhile, similar outputs from WRF-CMAQ simulation
provide complementary evidence for the fact that anthropogenic emissions exerted a
much stronger influence on $PM_{2.5}$ concentrations than meteorological conditions. In
addition to the combined effects of all emission-reduction measures, we further
employed WRF-CMAQ to quantify the relative contribution of different
emission-reduction measures to the decrease in $PM_{2.5}$ concentrations in Beijing from
2013 to 2017.

**4.2 The relative contribution of different emission-reduction measures to the decrease in PM$_{2.5}$ concentrations in Beijing**

The observed annual average PM$_{2.5}$ concentration in Beijing in 2017 was 58 mg/m$^3$, compared with 89.5 μg/m$^3$ in 2013. Based on WRF-CMAQ simulation, meteorological conditions contributed 6.7 μg/m$^3$ whilst the control of anthropogenic emissions contribute contributed 24.7 μg/m$^3$ to the total PM$_{2.5}$ reduction of 31.5 μg/m$^3$ in Beijing from 2013 to 2017. Specifically, local and regional emission-reduction accounted for 16.9 μg/m$^3$ and 7.8 μg/m$^3$ of PM$_{2.5}$ reduction. Local emissions and regional transport took up 68.4% and 31.6% of total anthropogenic emissions in Beijing. This result is consistent with our recent study (Chen et al., 2019). Chen et al. (2019) investigated four pollution episodes in Beijing in 2013, 2016, 2017 and 2018 respectively and found that local emissions accounted for 69.3%, 76.8%, 49.5% and 88.4% of total emissions in Beijing respectively. Except for the moderate pollution episode in 2017, local emissions caused more than two thirds of anthropogenic emissions in Beijing. Therefore, local emissions played a dominant role for PM$_{2.5}$ variations in Beijing in both long-term run and heavy pollution episodes. According to three emission-reduction scenarios designed, the regulation of coal boilers had the most significant effect on PM$_{2.5}$ reduction in Beijing and resulted in a decrease of 6.3 μg/m$^3$. Meanwhile, increasing clean fuels for residential use and industrial restructuring also exerted strong influences on PM$_{2.5}$ reduction and contributed to a decrease of 5.5 μg/m$^3$ and 3.4 μg/m$^3$ respectively. The three major strategies accounted for around half of the total effects of emission-reduction on PM$_{2.5}$ variations in Beijing.

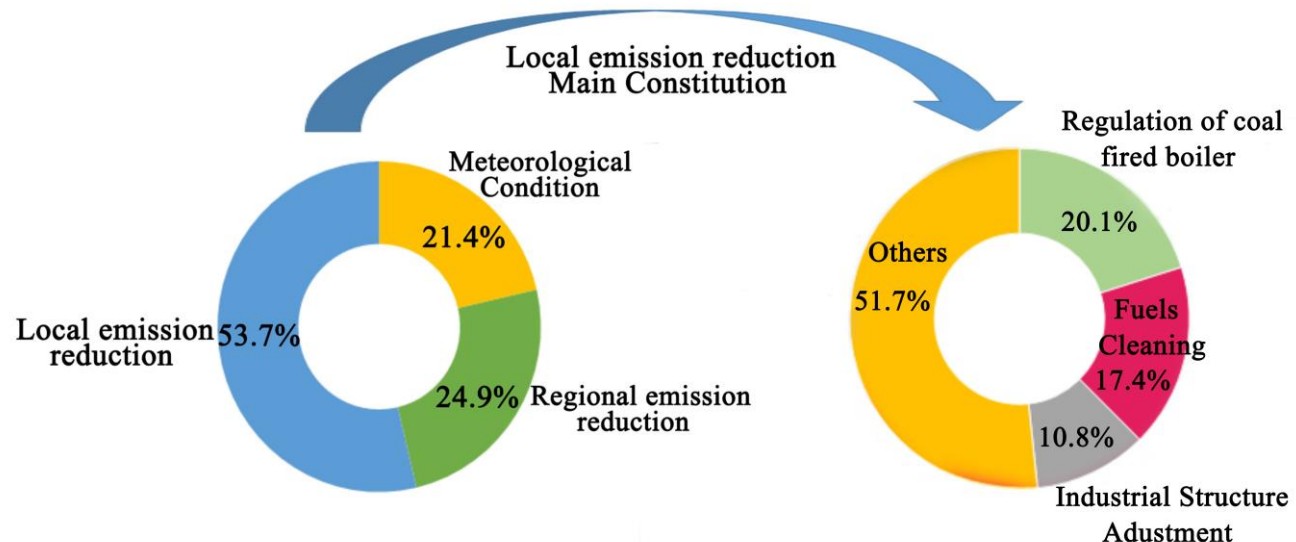

**Fig 5. The relative contribution of different influencing factors to the decrease of PM$_{2.5}$ concentrations in Beijing from 2013 to 2017**

**5 Discussion**

By the end of 2017, the Beijing Five-year Clean Air Action Plan (2013-2017) was completed and achieved its primary goal of reducing the annual average PM$_{2.5}$ concentration to less than 60 μg/m$^3$. Meanwhile, in November 2017, strong northerly winds in Beijing resulted in the cleanest winter in the past five years, raising arguments whether the favorable meteorological conditions were primarily responsible for PM$_{2.5}$ reduction or whether the significant improvement in air quality in Beijing was mainly attributed to the control of anthropogenic emissions. In this case, a quantitative comparison between the influence of meteorological conditions and emission-reduction on PM$_{2.5}$ reduction is necessary for comprehensively evaluating the Five-year Clean Air Action Plan. Based on two different approaches, this research revealed that the control of anthropogenic emissions contributed to around 80% of PM$_{2.5}$ reductions in Beijing from 2013 to 2017, indicating that the Five-Year Clean Air Plan exerted a dominant influence on air quality enhancement in Beijing. The large contribution of some specific emission-reduction measures may be obscured in the presence of favorable meteorological conditions. For instance, many residents may attribute the clean winter of 2017 to the notable strong winds without noticing some of major emission-reduction strategies implemented during this period.

A large-scale replacement of coal boilers with gas boilers was conducted in Beijing
and its neighboring areas since 2013. As quantified by WRF-CMAQ,the regulation of
coal boilers and increasing use of clean fuels for residential use jointly contributed to
an 11.8μg/m$^3$ decrease in $PM_{2.5}$ concentrations,much (almost twice) larger than the
6.7 μg/m$^3$ decrease caused by favorable meteorological conditions. In general,
although favorable meteorological conditions (e.g., strong winds) may lead to an
instant improvement of air quality, regular emission-reduction measures exert a
reliable and consistent influence on the long-term reduction of $PM_{2.5}$ concentrations in
Beijing. Given the satisfactory performance of the Five-year Clean Air Action Plan in
$PM_{2.5}$ reduction, such long-term clean air plan should be further designed and
implemented in Beijing and other mega cities with heavy $PM_{2.5}$ pollution.
Recently, with growing attention to the completion of the Five-year Clean Air Action
Plan, some other studies have also been conducted to evaluate this five-year plan.
Cheng, J. et al. (2019) employed a finer-scale and more detailed local
emission-inventory and quantified the relative contribution of multiple
emission-reduction strategies, including the control of coal-fired boilers, increasing
use of clean fuels, optimization of industrial structure, fugitive dust control, vehicle
emission control, improved end-of-pipe control, and integrated treatment of VOCs.
The relative contribution of these emission-reduction measures to $PM_{2.5}$ reduction in
Beijing from 2013 to 2017 was 18.7%, 16.8%, 10.2%, 7.3%, 6.0%, 5.7% and 0.6%
respectively. By contrast, our research revealed that three major emission-reduction
measures (the regulation of coal-fired boiler, increasing use of clean fuels and
industrial restructuring) contributed 20.1%, 17.4% and 10.8% of total $PM_{2.5}$ reduction
in Beijing from 2013 to 2017, which was very close to Cheng, J et al. (2019)'s
findings. Based on finer-scale local emission-inventories with more field-collected
emission data, Cheng, J et al. (2019) provided a comprehensive and reliable
understanding of the effects of multiple emission-reduction measures on $PM_{2.5}$
reduction in Beijing. The similar outputs from the two studies further proved the
reliability of WRF-CMAQ simulation. Meanwhile, Cheng, J et al. (2019) and UNEP
(2019) jointly quantified that the total amount of reduction in $SO_2$, $NO_x$, VOCs and
direct $PM_{2.5}$ induced by the control of anthropogenic emissions was 79420t, 93522t,
115752t and 44307t respectively, which was the major driver for the notable $PM_{2.5}$
reduction in Beijing from 2013 to 2017.
Although the "2+26" regional strategy for air quality improvement in Beijing has
become a hotly debated issue and growing emphasis has been placed on the proper
design and implementation of regional emission-reduction strategies in Beijing and its
surrounding cities,previous studies (Chen et al., 2019; Cheng, J. et al., 2019) and this
research proved that local emissions played a dominant role in affecting $PM_{2.5}$
concentrations in Beijing. Specifically, Chen et al. (2019) pointed out that with
intensive reduction of coal-fired boilers in Beijing-Tianjin-Hebei region, the relative
contribution of vehicle emissions to $PM_{2.5}$ concentrations in Beijing, especially during
heavy pollution episodes, could be up to 50%. To further improve air quality in
Beijing, stricter regulations on local vehicle emissions, including contingent strategies
during pollution episodes (e.g. odd-even license plate policy) and long-term policies
(e.g. increasing availability of public transit systems and electric cars) should be a
major priority for the next stage clean-air actions.
Based on KZ filtering, Cheng, N et al. (2019) and Ma et al. (2016) suggested the
seasonal component contributed dominantly to $O_3$ variations in Beijing. By
comparison, this research revealed that the short-term component contributed
dominantly to $PM_{2.5}$ variations in Beijing. These findings well explained the
phenomenon that ground ozone pollution in Beijing, controlled by seasonal variations
of emission and meteorological conditions (especially high-temperature and
low-humidity), simply occurred in summer, whilst $PM_{2.5}$ pollution in Beijing,
controlled by short-term variations of meteorological and emission factors, might
occur in all seasons. Consequently, contingent emission-reduction measures during
heavy pollution episodes are an effective approach to offset the short-term
deterioration of meteorological conditions and improve local air quality.
Despite the major contribution of emission-reduction measures to $PM_{2.5}$ reduction in
Beijing, meteorological influences, which contributed to 20% of $PM_{2.5}$ reduction,
should also be considered balancedly. In addition to the control of anthropogenic
emissions, $PM_{2.5}$ reduction may be realized through meteorological means. For the
winter of 2017, strong northwesterly winds led to instant improvement in air quality,
suggesting wind was a dominant meteorological factor for the accumulation or
dispersion of $PM_{2.5}$ in Beijing. Meanwhile, previous studies (Chen et al., 2017)
suggested that increasing wind speeds led to increased evaporation, increased
sunshine duration (SSD) and reduced humidity, which further reduced local $PM_{2.5}$
concentrations. In other words, strong winds help reduce $PM_{2.5}$ concentrations
through direct and indirect measures. In this light, the forthcoming Beijing
Wind-corridor Project, which includes five 500m-width corridors and more than ten
80m-width corridors to bring in stronger wintertime northwesterly winds, can be a
promising approach for promoting long-term favorable meteorological influences on
$PM_{2.5}$ reduction in Beijing.
**6 Conclusions**
To comprehensively evaluate the effect of the Beijing Five-year Clean Air Action Plan
(2013-2017), we quantified the relative contribution of meteorological conditions and
the control of anthropogenic emissions to the notable decrease in $PM_{2.5}$ concentrations
in Beijing from 2013 to 2017. Based on KZ filtering, we found that meteorological
conditions and emission-reduction accounted for 19.4% and 80.6% of the $PM_{2.5}$
reduction in Beijing, respectively. The large short-term component suggested that
short-term variations of meteorological and emission factors exerted a dominant
influence on the rapid variation of $PM_{2.5}$ concentrations in Beijing. Meanwhile,
WRF-CAMQ revealed that meteorological conditions and emission-reduction
contributed to 21.4% and 78.6% of $PM_{2.5}$ variations. Specifically, local and regional
emission-reduction measures contributed to 53.7% and 24.9% of $PM_{2.5}$ reduction. For
three major emission-reduction measures, the regulation of coal boilers, increasing
use of clean fuels for residential use and industrial restructuring contributed to 20.1 %,
17.4% and 10.8% of $PM_{2.5}$ reduction, respectively. Similar outputs from two models
suggested that the control of anthropogenic emissions contributed to around 80% of
the total decrease in $PM_{2.5}$ concentrations in Beijing from 2013 to 2017, indicating
that the Five-year Clean Air Plan worked effectively and such long-term clean air plan
should be continued in the following years to further reduce $PM_{2.5}$ concentrations in
Beijing.

**Acknowledgement**

Sincere gratitude goes to Tsinghua University, which produced the Multi-resolution Emission Inventory for China (http://meicmodel.org/) and Research center for air quality simulation and forecast, Chinese Research Academy of Environmental Sciences (http://106.38.83.6/), which supported the model simulation in this research. This research is supported by the National Key Research and Development Program of China (NO.2016YFA0600104) and National Natural Science Foundation of China (Grant Nos. 41601447).

**Author contribution**

Chen, Z., Gao, B. and Xu, B designed this research. Chen, Z wrote this manuscript. Chen, D., Zhuang, Y, Gao, B and Li, R. conducted data analysis. Chen, D and Zhuang, Y. produced the figures. Kwan, M., and Chen, B helped revise this manuscript.

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
