# Peer review of "The control of anthropogenic emissions contributed to 80% of the decrease in PM2.5 concentrations in Beijing from 2013 to 2017"

_Atmospheric Chemistry and Physics, 2018_

## Referee Comment (RC1) · Anonymous Referee #1 · 28 Feb 2019

The comment was uploaded in the form of a supplement: https://www.atmos-chem-phys-discuss.net/acp-2018-1112/acp-2018-1112-RC1-supplement.pdf

---

## Referee Comment (RC2) · Anonymous Referee #2 · 8 Apr 2019

It is really a hot topic for assessing the relative importance of meteorological parameters and emission reduction measures on the PM2.5 reduction from 2013. There is a similar manuscript on ACPD "Dominant role of emission reduction in PM2.5 air quality improvement in Beijing during 2013-2017: a model-based decomposition analysis". When compared with this one, different conclusions were drawn for the contribution of meteorological conditions to PM2.5 reduction in Beijing in from 2013 to 2017. However, this manuscript is far from the publishing criterion of ACP. I suggested rejection of this paper as the following reasons: (1) There are so many typesetting mistakes that I can not listed all of them. The authors could find the attached manuscript that I have labeled. Some mistakes indicated that the authors are not serious for the
scientific papers, such as the character subscript, the citation form of references. I am really confused why such kind of papers can be on the ACPD for open discussion. (2) The figures are made by Excel and in so poor quality, especially for Figure 2, 3 and 4. I really have a suspicion that are the authors know the quality of figures for scientific papers, not only say for ACP. (3) For the whole manuscript, it is just like a primary data analysis report, no discussion and no verification of the results. (4) I am quite disagree that at the last the authors wanted to assess the emission-reduction measures considering both PM2.5 and O3. They should know even for the assessing PM2.5 reduction, there existed large uncertainty especially for emission inventory, for subsector sources and for chemical speciations. More scientific questions should be addressed for improving the simulation. It suggested that the authors may be not quite sure about the research shortages on the emission inventory and its adoption on air quality modeling. (5) At last, I strongly suggest the authors carefully read the similar paper on ACPD and find the wide gap between yours and that one. In the future, the manuscripts should be carefully prepared. When you want to submit it to a high quality journal, please write it in a form of paper, not a report. Please also give the research shortages in science, not just say what you do.

Please also note the supplement to this comment:
https://www.atmos-chem-phys-discuss.net/acp-2018-1112/acp-2018-1112-RC2-supplement.pdf

———————————————————

---

## Author Response (AR1)

**To Editor:**

Dear Professor Yves Balkanski:

Thanks so much for providing us a chance to publish this manuscript in ACPD and receive valuable comments from reviewers. In the past four months, according to reviewers' comments, we have fully revised the manuscript in terms of literature review, language, figures and discussion. Many relevant references and more in-depth discussion have been added to the revised manuscript. More details concerning model descriptions, results and verification have been added as well. The introduction and discussion part has been fully rewritten. The revised manuscript has been enriched significantly. All figures have been reproduced. The English has been polished by some professional editors. Thanks again for processing our manuscript and providing us a chance to revise and resubmit this manuscript. Please feel free to contact us if additional revisions are required.

The very Best Ziyue To reviewer 1:

General comments: This manuscript quantified the relative contribution of meteorological conditions and emission control to the decrease of PM2.5 concentrations in Beijing using models. The results suggested that emission control was crucial for the air quality in Beijing, with a contribution of 80% to the decrease in PM2.5 concentrations using KZ filtering and WRF-CMAQ model. The topic is very interesting because the relative importance of influencing factors on air pollutants has been still unclarified. The method and result is helpful to understand the main influencing factors of air pollution and develop effective measures for pollution control and prevention in cities. I would thus recommend this manuscript to be published in ACP after improvement.

R: Thanks so much for your encouragement and useful suggestions. According to your comments, we have fully revised this manuscript. Please feel free to contact us if further revisions are required.

Specific comments:

Q1. L89-92. It is difficult to follow this sentence. Since DingLing Station and the MiYun Station as two background stations, what's purpose of choosing the Qianmen station and the Yufa Station? And the Yufa station can't be found in Fig.1.

R: Thanks so much for pointing this out. This is a good question. Beijing is a mega city with very large area. As a result, PM2.5 concentrations and emission factors vary across Beijing, and this is the reason why we selected these urban stations, as well as two background stations (Dingling and Miyun Station), Qianmen (Transport station with intense emissions), and the rural stations with distinct PM2.5 concentrations and emission factors. In this case, we can investigate that whether different

emission-scenarios influence the relative contribution of meteorological conditions to PM2.5 variations. And we are sorry that we did not make this figure clear to demonstrate the difference between different types of stations. The Figure 1 has been reproduced according to your comment in the revised manuscript and the Yufa station can be clearly identified in the updated Figure 1. Thanks so much for this.

Q2. L108. As far as I know, MECI emission inventory is only for 2012, 2014 and 2016, However in this paper, emission inventory in 2013 and 2017 were used. Please clarify more clearly.

R: This is a very good question. We are sorry that we did not make this clear. The Multi-resolution Emission Inventory for China, MEIC  $0.5^{\circ} \times 0.5^{\circ}$  emission inventory (http://www.meicmodel.org/) were updated annually. Therefore, the existing MEIC emission inventory are available from 2013 to 2016. Since 2017 MEIC is not available yet, we updated the 2016 MEIC emission inventory by considering the 2017 emission-reduction scenarios (e.g. the target of coal combustion reduction) required by the local government, a strategy that has been employed by previous studies ( Chen et al., 2019; etc. ). We are sorry that we did not make the explanation of emission inventory clear in the previous manuscript. And in the revised manuscript, we explained that the 2017 MEICInventory was updated from 2016 MEIC.

Chen, Z. et al. (2019) Evaluating the "2+26" Regional Strategy for Air Quality Improvement During Two Air Pollution Alerts in Beijing: variations of PM2.5 concentrations, source apportionment, and the relative contribution of local emission and regional transport. Atmospheric Chemistry and Physics 19(10):6879-6891.

Q3. L116. Which years' local environmental statistical data and reported emission data were used? From 2013 to 2017? Please clarify it. Did you compare the emission data of this Beijing local-emission inventory with others'? How about the difference? Since the

emission is a basic for your research.

R: Thanks so much for this. The local environmental statistical data used for this research were from 2013 to 2017. For your reference, we compared our statistical data with Annual report from National Environmental Statistics Bulletin (http://www.mee.gov.cn/gzfw\_13107/hjtj/qghjtjgb/) and the report from "2+26" center for air pollution prevention and control as follows. As you can see, the statistical data used for this research is highly consistent with other official data. The VOC value is very difficult to estimate and our data is very close to the data reported by the "2+26" center for air pollution prevention and control. Through this comparison, we believe the statistical data we collected for this research is valid for the following simulation analysis. Thanks again for pointing this out.

The comparison of The local environmental statistical data used for this research

and other official statistical data in 2017 (unit: 10k tons)

|                                                           | SO 2 | NO x | CO    | VOC   | NH 3 | PM 10 | PM 2.5 | BC   | OC   |
|-----------------------------------------------------------|-----------------|-----------------|-------|-------|-----------------|-------------------------|-------------------|------|------|
| Statistical data for this research                        | 1.38            | 10.15           | 49.54 | 13.47 | 3.20            | 14.74                   | 3.92              | 0.17 | 0.44 |
| National Environmental
Statistics Bulletin             | 1.38            | 12.16           | 52.03 | 24.24 | 3.26            | 14.68                   | 3.91              | 0.22 | 0.41 |
| "2+26" center for air pollution
prevention and control |                 | 9.24            | 48.98 | 13.93 | 3.16            | 13.82                   | 3.72              | 0.19 | 0.46 |

Q4. Section 3.1. Filtering is a key research method for this study, which decomposes the original signal into trend signal and seasonal signal and the disturbance. Although the contribution rate in Table 3 partially reflects the composition of the decomposition, a time series diagram is still necessary to show that the components are correct after filtering.

R: Thanks so much for this comment. Yes, the accuracy of KZ filtering is closely related to the reliability of our research and how to judge whether the result of KZ filtering is satisfactory. According to your comments, we again reviewed the general

principle of KZ filtering and other papers that employed KZ filtering, and found that no other criteria is supplied to verify KZ filtering. The commonly used criterion is that the closer the sum of three components to 1, the better filtering accuracy is. This is because a large value of the sum of the three components indicates that a majority of meteorological influences to PM2.5 variations have been considered in the KZ filtering. And for our research, the sum of three components for all stations are very close to 1, indicating a majority of meteorological influences has been filtered through KZ filtering. According to your comments, here we presented a KZ filtering curve as follows. Although we cannot easily judge the quality of KZ filtering according to the decomposed KZ components, we can see that the long-term component demonstrates a smooth curve whilst the trend of season component and short-term component is highly consistent with that of the original PM2.5 time series, especially for some simultaneous peaks. This means the extracted seasonal component and short-term component made a significant contribution to seasonal and short-term variations of original PM2.5 concentrations in Beijing from 2013 to 2017, indicating a satisfactory KZ filtering result. Thanks again for pointing this out and we have added relevant explanation to the revised manuscript.

Long-term, seasonal and short-term component extracted from original PM2.5 time series in Beijing from 2013 to 2017 using KZ filtering.

Q5. Section 3.2.2. Model evaluation is the key point in this paper. If the model data is not consistent with observation, contribution of emission control is out of the question. It seems that lots of data are far from the observation especially during the heavy air pollution days. So it is better to convert Fig 2 to time series plots, which can tell us more detailed information about the model evaluation.

R: Thanks so much for this comment. Generally, model-simulation cannot perfectly fit the actual curve of PM2.5 concentrations due to the deficiency of emission-inventories, the incomplete descriptions of reactions mechanisms for secondary formation of PM2.5 and other uncertainties. For long-term simulation based on unified parameter setting, the model simulation outputs can demonstrate notable difference with the observed PM2.5 concentrations, especially during heavy pollution episodes (Li et al., 2011). This is because commonly employed CTMs do not fully consider heterogeneous/aqueous reactions, which significantly deteriorate PM2.5 pollution (Chen et al., 2016). So yes, you are right. It is a common challenge that long-term CTM simulation may significantly underestimate PM2.5 concentrations during heavy pollution episodes. And this is also the situation of our simulation. However, despite the relative large bias during the heavy pollution episodes, the general simulation accuracy: the correlation coefficient R, normalized mean bias (NMB), normalized mean error (NME), mean fractional bias (MFB) and mean fractional error (MFE) between observed and simulated data was 0.69~0.74, 11%~17%, 20%~27%, -21%~17%, and 15%~27% respectively is satisfactory (EPA, 2005; Boylan et al., 2006). Following your constructions, we have convert the Fig 2 to time series plots. In addition, we acknowledged that the model simulation produced some large bias during heavy pollution episodes, caused by the common limitations of CTMs.

Li, G., Zavala, M., Lei, W., Tsimpidi, A.P., Karydis, V.A., Pandis, S.N., Canagaratna, M.R., Molina, L.T., 2011. Simulations of organic aerosol concentrations in Mexico City using the WRF-CHEM model during the MCMA-2006/MILAGRO campaign. Atmos. Chem. Phys., 11: 3789-3809.

Chen, D., Liu, Z., Fast, J., Ban, J. 2016. Simulations of sulfate-nitrate-ammonium (sna) aerosols during the extreme haze events over northern china in october 2014. Atmospheric Chemistry and Physics, 16(16), 10707-10724.

Q6. Section 3.2.2. You verified the accuracy of the WRF-CMAQ model using the data of three stations. How about other urban stations? This does not mean that the figures of all

stations should be supplemented, but it does require that extrapolation to difference between observed and WRF-CMAQ simulated PM2.5 concentrations.

R: Thanks so much for this comment. In the previous manuscript, we verified the data of three stations and following your comments, we further added the verification of another three urban stations to the revised manuscript to prove the reliability of the model simulation. Thanks again for this valuable comment.

Q7. L339. How did you get the conclusion "KZ filtering provides a more reliable method"? Just because the KZ filtering was station-based and WRF-CMAQ model was a grid-based? The averaged relative contribution of meteorological variations to PM2.5 reduction using the WRF-CMAQ model was very similar to that using KZ filtering. Verification is very important for the model results. So what's the criteria for judging reliability of your model?

R: This is a very good question. The advantage of KZ filter is that this statistical method is based on the observed meteorological data and PM2.5 concentrations and predicts the variations of airborne pollutants on the hypothesis of unchanged meteorological conditions. In this case, by comparing the original and filtered time series of airborne pollutants, the relative contribution of meteorological conditions to long-term variations of airborne pollutants. Since KZ filtering is based on observed data, and simply consider the influence of time-series meteorology data on PM2.5 time series, less uncertainty is involved in this analysis, KZ is influenced mainly by the variations of meteorology-PM2.5 interactions in different areas and seasons. On the other hand, CTMs, e.g. WRF-CMAQ or WRF-CAMx considers both meteorological conditions (which is large-scale meteorological data, not as

accurate as local observed meteorological data) and anthropogenic emissions for estimating PM2.5 concentrations under different emission scenarios. Therefore, the accuracy of these models are not only decided by proper understanding of meteorological data, but also the reliability of emission inventories and proper descriptions of reaction mechanisms for PM2.5 production, especially during heavy pollution episodes, which is a major challenge for current model simulation. For instance, without consideration of heterogeneous/aqueous reactions between sulfate, nitrate, and ammonium (denoted as SNA) in high-humidity environment, WRF-CAMx failed to simulate maximum PM2.5 concentrations during extreme haze episodes (Chen, D. et al., 2016). And the emission inventories, no matter how fine they produced, are quite different from actual emission situations. Therefore, model simulated PM2.5 concentrations, especially the relative contribution of anthropogenic emissions to PM2.5 concentrations, are influenced by much more factors than the KZ filters. In this case, KZ filtering is most suitable for quantifying the relative contribution of meteorological conditions to long-term variations of airborne pollutants and recently been increasingly employed for this type of research.

On the other hand, since the emission inventory includes different emission sources, CTMs, e.g. WRF-CMAQ or WRF-CAMx, are suitable for quantifying the relative contribution of different sources to PM2.5 variations, though large variations remained.

Q8. L398-399. Supplement the correlation coefficient between wind speed and PM2.5. And how about the influence of the other meteorological parameters (such as T, RH, wind direction) on PM2.5?

R: As detailed explained in our previous studies (Chen, Z et al., 2017; 2018), the

causal influence of individual meteorological factors on PM2.5 concentrations cannot be precisely quantified using correlation analysis, as the complicated interactions between different meteorological factors. Instead, a robust model CCM, which can remove the influence of other influencing meteorological factors, has been employed in our research to extract the dominant meteorological factors for PM2.5 concentrations in Beijing and other mega cities across China. The p value, similar to correlation coefficient, is a quantitative and more reliable indicator of meteorological influence on PM2.5 concentrations. Detailed information concerning the influence of many meteorological factors on PM2.5 concentrations can be found in our research (Chen, Z. et al., 2017; 2018). Here, we listed part of the table here as below for your reference. Meteorological influences on PM2.5 concentrations vary across seasons, and SSD (sunshine duration), wind speed, humidity and temperature are major influencing factors for PM2.5 concentrations in Beijing and other mega cities in China, especially the North China plain. Meanwhile, wind direction was not significantly correlated with PM2.5 concentration in Beijing ( Chen et al., 2017, 2018). The major reason is that the influence of wind direction on PM2.5 concentrations is subjected to geographical conditions and not strongly correlated to PM2.5 conditions (Chen et al., 2017).

Therefore, for this research, based on our previous studies on PM2.5-meteorology interactions in Beijing, the major meteorological factors temperature, humidity, wind speed and solar radiation are used for the KZ filtering.

Chen, Z.Y., Cai, J., Gao, B.B., Xu, B., Dai, S., He, B., Xie, X.M., 2017. Detecting the causality influence of individual meteorological factors on local PM2.5 concentrations in the Jing-Jin-Ji region, Scientific Reports, 7:40735.

Chen, Z.Y., Xie, X., Cai, J., Chen, D., Gao, B., He, B., Cheng, N., Xu, B., 2018.

Understanding meteorological influences on PM2.5 concentrations across China: a temporal and spatial perspective, Atmos. Chem. Phys, 18: 5343-5358

The correlation coefficient and  $\rho$  value of different meteorological factors (Temperature, humidity and wind speed) on seasonal PM2.5 concentrations in Beijing (Chen et al., 2017)

| City    | Spring       | Summer          | Autumn          | Winter          |  |
|---------|--------------|-----------------|-----------------|-----------------|--|
|         | RHU**        | RHU**           | RHU**           | RHU**           |  |
|         | (0.532,      | (0.648, 0.546)  | (0.587, 0.555), | (0.738,0.738),  |  |
|         | 0.490) SSD** |                 | SSD**           | SSD**           |  |
| Delline |              | (-0.447,0.324)  | (-0.509,        | (-0.715,0.577), |  |
| Beijing |              | TEM**           | 0.410),         | WIN**           |  |
|         |              | (0.554, 0.455), | WIN**           | (-0.558, 0.531) |  |
|         |              |                 |                 |                 |  |

(-0.468,0.223),

\*\*Correlation is significant at the 0.01 level (2 tailed); \*Correlation is significant at the 0.05 level (2 tailed).

The first value in the brackets presents the correlation coefficient between the meteorological factor and PM2.5 concentration.

The second value presents the quantitative influence of individual meteorological factors on local PM2.5 concentration ( $\rho$  value), whilst the feedback effects of PM2.5 on these meteorological factors are not listed here.

NA indicates that no significant correlation exists between the meteorological factor and PM2.5 concentration.

Q9. Contribution of local emission-reduction measures was discussed in this paper. Please

describe the reduction amount of each pollutant (SO2, NOx, PM) from each measure (e.g. coal boilers, residential use, industrial restructuring). It is better to discuss the contribution of reduction of different pollutants, which could reflect the effect of primary emission and secondary formation.

R: Thanks so much for this suggestion. In the past several months, including our research, there are some recent publications to discuss the variations of emission factors from 2013 to 2017 in Beijing and the underlying drivers for this. Specifically, UN published a formal report on air pollution in Beijing in the past two decades and released some official statistical data for the emissions of different pollutants in Beijing from 2013 to 2017. Therefore, in the revised manuscript, we fully reviewed these relevant studies and conducted an in-depth discussion on how emission-reduction measures have changed the relative contribution of different sources to PM2.5 concentrations in Beijing. Thanks again for this valuable comment, which improved this manuscript significantly.

Technical corrections:

L40. I suggest that authors change keyword "PM2.5 reduction" to "PM2.5".

**R: Corrected.**

L144. Supplement the time period for "a historical record".

R: Actually, there are multiple haze episodes in December 2012 and January 2013, and the historical high record was observed during these episodes, no specific time period given by previous studies (e.g. Zhang, R., 2013).

L184. Supplement the link for "the website PM25.in".

**R: Corrected.** P183. Check and revise the Formula (9).

R: Corrected.

**Reviewer 2:**

It is really a hot topic for assessing the relative importance of meteorological parameters and emission reduction measures on the PM2.5 reduction from 2013. There is a similar manuscript on ACPD "Dominant role of emission reduction in PM2.5 air quality improvement in Beijing during 2013-2017: a model-based decomposition analysis". When compared with this one, different conclusions were drawn for the contribution of meteorological conditions to PM2.5 reduction in Beijing in from 2013 to 2017. However, this manuscript is far from the publishing criterion of ACP.

I suggested rejection of this paper as the following reasons:

**To Referee 2:**

Thanks so much for your general and detailed comments on our manuscript. We have fully revised this manuscript according to your general and detailed comments mentioned in this and the complementary file. Specifically, we really appreciate you mentioned a relevant paper, which also investigate the relative contribution of meteorological conditions and anthropogenic emissions to PM2.5 variations in Beijing from 2013 to 2017. We also noticed this paper and actually very glad to see that this research and our research revealed the same fact that anthropogenic emissions contributed majorly to PM2.5 variations in Beijing from 2013 to 2017 from different perspectives. Cheng et al. (2019)'s research employed more fine-scale emission-inventories to specifically quantify a diversity of emission sources to PM2.5 concentrations in Beijing from 2013 to 2017. Meanwhile, the major aim of this research is to use the statistical model KZ to filter the influence of meteorological variations and also use CTM model to verify the result from the KZ model. To better present our results, we have fully revised our manuscript to include more discussion of these relevant studies and recently released reports, highlight the unique contribute of Cheng et al. (2019) and our research, and conclude the combined theoretical and practical

significance of these studies to air quality improvement in Beijing and other mega cities in China. As a result, we do believe a significantly improved version of our manuscript and Cheng et al. (2019)'s research can jointly contribute a lot to the ACP society.

Meanwhile, although the authors have already published many top journals including Science, The Lancet, PNAS and of course some ACP papers, clearly the figures, text and languages of this manuscript can be improved a lot, especially according to your comments. Thanks again for giving us suggestions to improve the presentation of this manuscript. We have carefully reproduced all these figures and polished the structure and language according to your comments. We are more than willing to conduct further revisions if additional requirements are given.

Thank again for reviewing our manuscript and your valuable comments indeed help us a lot.

(1) There are so many typesetting mistakes that I can not listed all of them. The authors could find the attached manuscript that I have labeled. Some mistakes indicated that the authors are not serious for the scientific papers, such as the character subscript, the citation form of references. I am really confused why such kind of papers can be on the ACPD for open discussion.

R: Thanks a lot for your comment. Although we have already published several papers in ACP, clearly there is room for us to improve this manuscript. And the forthcoming of more qualified ACP paper do rely on more and more strict requirements on received manuscripts. We are very grateful that the Co-Editor and the reviewers rated high on this manuscript and accepted this manuscript for ACP discussions, and thus we can receive highly valuable comments from qualified and strict experts like you. Thanks again for all your comments in the revised manuscript. We have fully revised this manuscript according to your comments listed in your attached files.

(2) The figures are made by Excel and in so poor quality, especially for Figure 2, 3 and 4. I really have a suspicion that are the authors know the quality of figures for scientific papers, not only say for ACP.

R: Thanks so much for your comment. Actually, some figures are not produced by Excel and we are very sorry that you think these figures are in poor quality and do not like them. As mentioned above, since we have published many high-level papers and previous reviewers, including other three reviewers for this ACPD manuscript, do not question the quality of these figures, therefore we do not have a stricter standard for figure production. Thanks again for pointing this out. We reproduced all these figures and hope the reproduced figures can better fit your requirements. Please feel free to let us know if you have additional requirements for these figures and we are more than willing to reproduce them again fully according to your suggestions.

(3) For the whole manuscript, it is just like a primary data analysis report, no discussion and no verification of the results.

R: Thanks so much for this comment. According to your comments, we have fully revised the manuscript in the introduction, discussion and result part to highlight the practical meaning and the correlation between this research and relevant studies. Actually, model simulation for three sites has already been there in the previous manuscript. According to the suggestions of you and another reviewer, we added the verification of additional three sites. We are sorry that we did not make this clear and have added more explanation of the verification and potential simulation error to the revised manuscript. Thanks again for your valuable suggestions. (4) I am quite disagree that at the last the authors wanted to assess the emission-reduction measures considering both PM2.5 and O3. They should know even for the assessing PM2.5 reduction, there existed large uncertainty especially for emission inventory, for subsector sources and for chemical speciations. More scientific questions should be addressed for improving the simulation. It suggested that the authors may be not quite sure about the research shortages on the emission inventory and its adoption on air quality modeling.

R: Thanks so much for pointing this out. We are sorry that we did not make it clear. We are not saying that we would like to use CTMs to assess the effects of emission-reduction measures on ozone and PM2.5 reduction. We understand a diversity of uncertainties related to CTM simulations. Actually, the negative, positive or inconsistent effects of emission-reduction measures on PM2.5 and ozone concentrations can be understood simply based on the observation data. According to our previous studies, we found ozone concentrations in Beijing were even enhanced (based on observed data) while specific emission-reduction measures for PM2.5 reduction were conducted. This fact is also proved by some relevant studies based on observation data that proved ozone concentrations were not consistently reduced during specific events (e.g. 2014 APEC meeting) with emission-reduction measures. That's the reason we mentioned that emission-reduction measures for PM2.5 concentrations may not effectively reduce ozone concentrations and emission-reduction measures should be balancedly considered for PM2.5 and ozone pollution. To avoid unnecessary confusions, we have fully revised the discussion part by including more details on the introduction of relevant studies and removing the discussion of ozone pollution management. Thanks again for your comment.

(5) At last, I strongly suggest the authors carefully read the similar paper on ACPD and find the wide gap between yours and that one. In the future, the manuscripts should be carefully prepared. When you want to submit it to a high quality journal, please write it in a form of paper, not a report. Please also give the research shortages in science, not just say what you do.

R: Thanks again for recommending this manuscript, which is a well-presented work and quantified the contribution of different sources to PM2.5 reduction from the model-simulated perspective. As we know, CTMs are affected by a diversity of uncertainties, including the variations in PM2.5-meteorology interactions, emission inventories, incomplete descriptions of reaction mechanisms between precursors under heavy pollution episodes, difficulties in parameter setting for long-term running and so forth. In this case, the statistical model, KZ employed to filter meteorological influences based on observed time series of PM2.5 concentrations and meteorological conditions are affected by many less influencing facts to quantify the relative contribution of meteorological conditions and anthropogenic emissions. So, based on your detailed and valuable comments, and a careful study of relevant high-level publications, we do believe that we can properly revise this manuscript and improve its quality significantly. A significantly improved version of our manuscript and Cheng et al. (2019)'s research can jointly contribute a lot to a comprehensive understanding of anthropogenic and meteorological influences on PM2.5 reduction from 2013 to 2017. Thanks again for recommending this manuscript and all these valuable comments on our manuscript.

1The control of anthropogenic emissions contributed to 80% of the2decrease in PM2.5 concentrations in Beijing from 2013 to 2017

3 Ziyue Chen1,2, Danlu Chen1, Mei-Po Kwan Meipo Kwan3,43, Bin Chen4 Chen5, Bingbo

4 Gao5Gao6\*, Yan Zhuang1, Ruiyuan Li1, Bing <del>Xu6Xu7\*</del>

5 1College of Global and Earth System Science, Beijing Normal University, 19 Xinjiekou

6 Street, Haidian, Beijing 100875, China.

7 2Joint Center for Global Change Studies, Beijing 100875, China.

8 3Department of Geography and Geographic Information Science, University of Illinois at

9 Urbana-Champaign, Urbana, IL 61801, USA.

10 4Department of Human Geography and Spatial Planning, Utrecht University, 3584 CB Utrecht,

11 The Netherlands

4Department 5Department of Land, Air and Water Resources, University of California, Davis, CA
 95616, USA

14 College of Land Science and Technology, China Agriculture University, Tsinghua East Road,

15 Haidian District, 100083, China.

16 27Ministry of Education Key Laboratory for Earth System Modeling, Department of Earth System

17 Science, Tsinghua University, Beijing 100084, China

18 \*To whom correspondence should be addressed. Email: gaobb@lreis.ac.cn or
19 bingxu@tsinghua.edu.cn

**20 Abstract**

With the completion of the Beijing Five-year Clean Air Action Plan by the end of 21 22 2017, the annual mean PM2.5 concentrations in Beijing dropped dramatically to 58.0  $\mu g/m^3$  in 2017 from 89.5  $\mu g/m^3$  in 2013. However, controversies exist to argue that 23 24 favorable meteorological conditions in 2017 that helped pollution dispersion-were the 25 major factor driver for such rapid decrease in PM2.5 concentrations. To 26 comprehensively evaluate this five-year plan, we employed Kolmogorov-Zurbenko 27 (KZ) filtering and-\_a-WRF-CMAQ model-to quantify the relative contribution of meteorological conditions and the control of anthropogenic emissions to PM2.5 28 29 reduction in Beijing from 2013 to 2017. For these five years, the relative contribution 30 of emission-reduction measures to the decrease of PM2.5 concentrations in Beijing

31 calculated by KZ filtering and the-WRF-CMAQ model-was 80.6% and 78.6%

**带格式的:**字体:14磅 **带格式的:**段落间距段前:0磅,段后:6磅 32 respectively. KZ filtering suggested that short-term variations of meteorological and 33 emission conditions contributed majorly to rapid changes of PM2.5 concentrations in Beijing. - The-WRF-CMAQ model further-revealed that the relative contribution of 34 local and regional emission-reduction-measures contributed toto 53.7% and 24.9% of 35 36 the PM2.5 reduction decrease in Beijing was 53.7% and 24.9% respectively. For local 37 emission-reduction measures, the regulation of coal boilers, increasing use of clean fuels for residential use, and industrial restructuring, the regulation of raise dust and 38 39 vehicle emissions contributed to 20.1 %, 17.4% and, 10.8%, 3.0% and 2.4% of PM2.5 40 reduction respectively. Both models suggested that the control of anthropogenic 41 emissions contributed to accounted for around 80% of the total decrease in PM2.5 42 concentrations reduction in Beijing, indicating that emission control reduction was crucial for the notable improvement in air quality enhancement in Beijing from 2013 43 44 to 2017. Therefore, Consequently, such long-term air quality clean plan should be continued infor the future following years to further reduce PM2.5 concentrations in 45 Beijing. Considering that different emission reduction measures exert distinct effects 46 47 on PM2.5 reduction and existing emission reduction measures work poorly to reduce 48 ozone concentrations, future strategies for emission reduction should be designed and 49 implemented accordingly. 50 Keywords: PM2.5, anthropogenic emissions, meteorological conditions,

Kolmogorov-Zurbenko (KZ) filtering, WRF-CMAQ

51

**52 1 Introduction**

53 In December 2012 January 2013, a heavy persistent haze episodes occurred in Beijing, during which the highest hourly PM2.5 concentrations once reached-\_886  $\mu$  g/m3, a historic high 54 record, a historical record., The extremely Hhigh-concentration, PM2.5 concentrations led to 55 long-lasting black and thick fogs, which not only significantly influenced people's daily life 56 57 (low-visibility induced traffic jam), but also exerted strong negative influences on posed a severe threat to public health (Brunekreef et al., 2002; Dominici et al., 2014; Nel et al., 2005; 58 59 Zhang et al., 2012; Qiao et al., 2014). Since then, severe haze haze episodes have frequently occurred been observed in Beijing and other regions in across China (Chan et al., 2008; 60 61 Huang, R., et al., 2014; Guo et al., 2014; Zheng et al., 2015), and PM2.5 pollution has become 62 one of the most concerned environmental issues in China. Since 2013Consequently, a 63 national network of ground stations for monitoring hourly  $PM_{2.5}$  concentrations has been established gradually, including 35 ground observation stations in Beijing, which provide 64 65 important support for proper management and in depth investigation of better understanding and managing PM2.5 concentrations. Meanwhile, for To effectively reducing mitigate local 66 67 PM2.5 pollutionconcentrations, Beijing Municipal Government the local government proposedreleased-the "Beijing Five-year Clean Air Action Plan" (2013-2017)". This plan 68 69 suggested the specific aim that the annual mean PM2.5 concentrations in Beijing should be 70 reduced from 89.5 µg/m3-in 2013 to 60 µg/m3-in 2017 and with included a series of long-term emission-reduction measures, including shutting down heavily polluting factories, 71 72 restricting traffic emissions and replacing coal fuels with clean energies, and. Furthermore, 73 for reducing high PM2.5 concentrations during severe haze episodes, Beijing Municipal 74 Government published \_\_the "Heavy Air Pollution Contingency Plan" in 2012, and further 75 revised this plan in March 2015. According to this plan, with a series of contingent emissionreduction measures during heavy pollution episodes should be implemented according to the 76 77 severeness of PM2.5 pollution episodes. By the end of 2017, these long-term and contingent 78 emission-reduction measures had-worked together-jointly to reduce the annually mean PM2.5 79 concentration in Beijing from 89.5 µg/m3 in 2013 to 58.0 µg/m3 in 2017, indicating a great success of PM2.5 management during the past five years. 80

81 82 In addition to anthropogenic emissions, the strong meteorological influences on  $PM_{2.5}$  concentrations in Beijing have been widely acknowledged (Cheng et al., 2017; Chen<del>, Z.</del> et

|-------|-----|----|---|
|       |     |    |   |

83 al., 2016, 2017, 2018; UNEP, 2016; Wang et al., 2014; Zhao et al., 2013). For instance, Chen, 84 Z et al. (2016) found that for 2014, more than 180 days in Beijing experienced a dramatic daily AQI (Air Quality Index) change ( $\triangle$ AQI>50) (Chen, Z. et al., 2016), compared with 85 the previous day. Considering that anthropogenic emissions the total emission of airborne 86 87 pollutants for a mega city hardly unlikely changed significantly on a daily basis, the rapid variations of meteorological conditions in Beijing werewas one important-major driver for 88 89 the dramatic change of daily air quality in Beijing. In winter 2017, strong northwest winds 90 led to-a favorable meteorological conditions for PM2.5 diffusion and low PM2.5 concentrations in Beijing, which. This In this case, there arise raiseds the controversy that 91 92 meteorological conditionsy, instead of emission-reduction measures, made a major 93 contribution toaccounted for the remarkable PM2.5 reduction of PM2.5 concentrations in 94 Beijing from 2013 to 2017. In this case, wWith the completion of the five-year plan, it is highly necessary to quantify the relative contribution of meteorological conditions and 95 96 emission-reduction measures to the notable remarkable decrease inof PM2.5 concentrations in 97 Beijing from 2013 to 2017.

98 In recent years, growing studies have been conducted to investigate meteorological and 99 anthropogenic influences on long-term PM2.5 variations. Based on Goddard Earth Observing 100 System (GEOS) chemical transport model (GEOS-Chem), Yang et al (2016) revealed that 101 the relative contribution of meteorological conditions to PM2.5 variations in Eastern China 102 from 1985 to 2005 was 12% from 1985 to 2005. Based on a multiple general linear model 103 (GLM), Gui et al. (2019) quantified that meteorological conditions accounted for 48% of 104 PM2.5 variations in Eastern China from 1998 to 2016. Through a two-stage hierarchical 105 clustering method, Zhang et al. (2018) calculated that the relative contribution of 106 meteorological conditions to heavy pollution episodes within the Beijing-Tianjin-Hebei 107 region was larger than 50% from 2013 to 2017. These studies quantified the overall 108 meteorological influences on long-term PM2.5 variations using different statistical models 109 and chemical transport models (CTMs). However, due to strong interactions between 110 individual meteorological factors, traditional statistical methods such as correlation analysis 111 and linear regression may be biased significantly when quantifying meteorological 112 influences on PM2.5 concentrations (Chen et al., 2017). On the other hand, the accuracy of 113 CTMs can be influenced largely by the uncertainty in emission inventories (Xu et al., 2016) 114 and deficiency of heterogeneous/aqueous processes (Li, G. et al., 2011). Therefore,

|-------|-----|----|---|

| 115 | multiple advanced models should be comprehensively considered employed to better quantify     |
|-----|-----------------------------------------------------------------------------------------------|
| 116 | meteorological influences on PM 2.5 concentrations (Pearce, J et al., 2011).       |
| 117 | To evaluate this fiveyear's clean-air plan, we employ an advanced statistical model,          |
| 118 | Kolmogorov-Zurbenko (KZ) filtering, which is advantageous of filtering meteorological         |
| 119 | influences on long-term time series of airborne pollutants, and a CTM model, WRF-CMAQ,        |
| 120 | which is advantageous of quantifying the relative contribution of different emission sources, |
| 121 | to comprehensively investigate the relative contribution of meteorological conditions and     |
| 122 | emission-reduction to PM 2.5 reduction in Beijing from 2013 to 2017 respectively.  |
| 123 | To this end, we employ different approaches in this paper to comprehensively estimate         |
| 124 | adjusted PM2.5 concentrations in Beijing while eliminating the influence from the variation   |
| 1   |                                                                                               |

125 in meteorological conditions and thus quantify the relative contribution of 126 emission reduction measures to the decrease of  $PM_{2.5}$  concentrations. In this light, this 127 research provides important insight for better designing and implementing successive clean 128 air plans in the future to further mitigate  $PM_{2.5}$  pollution in Beijing.

**129 2 Data Sources**

**130 2.1 PM2.5 and meteorological data**

131 In this study, hourly PM2.5 concentration data were acquired from the website PM25.in (www.PM25.in), which collects official data provided by China National Environmental 132 133 Monitoring Center (CNEMC). Beijing has established an advanced air quality monitoring 134 network with 35 ground stations across the city. Considering the major contribution of industry and traffic-induced emissions in urban areas, we selected all twelve urban stations 135 136 to analyze spatio-temporal variations of PM2.5 concentrations and quantify their influencing 137 factors. In addition to these urban stations, we selected two background stations, the 138 DingLing Station located in the suburb and the MiYun Reservoir Station located in the outer 139 suburb, one transportation station (the Qianmen station) located close to a main road, and 140 one rural station (the Yufa Station) which that is far away from central Beijing for the 141 following analysis. The DingLing and MiYun Reservoir Station were chosen as background 142 stations by the Ministry of Environmental Protection of China. These two stations receive 143 limited influence from anthropogenic emissions due to their location in suburban and outer

144 suburban areas. The Qianmen transportation station received more influences from vehicle emissions. Comparing the Long-term variations in of PM2.5 concentrations and its 145 anthropogenic and meteorological driving factors-in different type of stations provides a 146 147 useful reference for comprehensively understanding the effects of emission-reduction 148 measures on the reduction of PM2.5 decreaseconcentrations in Beijing from 2013 to 2017in the past five years. Meteorological data for this research were collected from the 149 Guanxiangtai Station (GXT,54511, 116.46° E, 39.80° N), Beijing and were downloaded from 150 151 the Department of Atmospheric Science, College of Engineering, University of Wyoming 152 (http://weather.uwyo.edu/upperair/sounding.html). Both the-PM2.5 and meteorological data 153 were collected from January 1st, 2013 to December 31st, 2017. The locations of these 154 selected stations are shown in Fig 1. Meteorological data for this research were collected the Guanxiangtai Station (GXT,54511, 116 160 155 from 20 800 156 downloaded from the Department College 157 University of Wyoming (http://weath Poth the PM and meteorological data were collected from January 1er, 2013 to December 31 158

Fig 1. Locations of different ground monitoring stations.

**161 **2.2 Emission inventories**

173

174

175

\_

162 For this research, we employed both regional and local emission inventories for running simulation. Multi-resolution Emission Inventory for 163 model China, MEIC, (http://meicmodel.org/) provided by Tsinghua University, were employed as the regional 164 165 emission inventories. MEIC has been widely employed and verified as a reliable emission inventory by a diversity of studies (Hong et al., 2017; Saikawa et al., 2017; Zhou et al., 2017; 166 167 etc.). For simulating five-year PM2.5 concentrations, MEIC from 2013 to 2017 are required. 168 Since official MEIC 2017 washas not been available yet, we employed a strategy from 169 previous studies (Chen et al., 2019; etc) and updated-the MEIC 2016 for simulating 170 emission-reduction scenarios and  $PM_{2.5}$  concentrations in 2017 by considering official the 2017 emission-reduction plansscenarios (e.g. the target of coal combustion reduction) 171 172 required by the local government.

176 Different from regional emission inventories, local emission inventories are usually 177 produced independently by local institutiones. The Beijing local-emission inventoryies 178 employed for this research is-wereas produced and updated by Beijing Municipal Research 179 Institute of Environmental protection, -fully according to the requirement of MEP on the 180 production of local emission inventories within the-Beijing-Tianjin-Hebei region. This 181 Beijing local-emission inventory from 2013 to 2017 wasis produced by synthesizing local 182 environmental statistical data and reported emission data, carrying out field investigations 183 and conducting a series of estimation according to Beijing Five-year Clean Air Action Plan. 184 This Beijing local emission inventory It is highly consistent with other official statistical data, 185 such as the Annual report from National Environmental Statistics Bulletin 186 (http://www.mee.gov.cn/gzfw 13107/hjtj/gghjtjgb/), and has been formally employed for the 187 implementation of recent "2017 Air Pollution Prevention and Management Plan for the 188 Beijing-Tianjin-Hebei Region and its Surrounding Areas" (MEP, 2017).\_

**189 3 Methods**

A key step for quantifying the relative contribution of anthropogenic emissions to the decrease of PM2.5 concentrations-variations is to properly filter meteorological influences on PM2.5 concentrations, which is highly challenging and rarely investigated by previous studies. Therefore, we employed both a statistical method and a chemical transport modeCTM1 in this study to comprehensively evaluate the role of anthropogenic emissions and meteorological conditions in the decrease of PM2.5 concentrations in Beijing <del>during the past</del> five yearsfrom 2013 to 2017.

**197 3.1 Kolmogorov-Zurbenko (KZ) filtering**

198 Since meteorological conditions exert a strong influence on PM2.5 concentrations in Beijing, 199 the removal of seasonal signals from time series of meteorological factors results inproduces 200 data sets suitable for understanding the trend of PM2.5 concentrations mainly influenced by 201 anthropogenipic factors (Eskridge et al., 1997). To better analyze the trend of time series data 202 without the disturbances from large variations of other major influencing variables, a 203 statistical method-called Kolmogorov-Zurbenko (KZ) filtering was proposed by Rao et al. 204 (1994). The KZ filter is advantageous in-of removing high-frequency variations in the-data 205 sets based on thethrough iterative moving average. Eskridge et al. (1997) compared four 206 major approaches for trend detection, including PEST, anomalies, wavelet transform, and the 207 KZ filter, and suggested that KZ achieved higher the confidence in detecting long-term trend 208 of the KZ filter was much higher than that of the other methods models. Due to its reliable 209 performance in trend detection in complicated ecosystems, the KZ filter has frequently been 210 increasingly employed to remove seasonal signals of meteorological conditions and extract 211 long-term trend of airborne pollutants (Zurbenko, et al., 1996; Eskridge, et al., 1997; Kang, 212 et al., 2013; Ma et al., 2016; Cheng, N -et al., 2019). One potential limitation of the KZ 213 filter is that iterative moving average (m) may impose an influence on detecting abrupt 214 changes of variations. Therefore, Zurbenko et al. (1996) proposed an enhanced KZ filter that 215 employed a dynamic variable *m* that decreases decreased with the increase in changing rate. 216 For this research, we, which is employed this dynamic m in this study to estimate the produce 217 an adjusted time-series of modified PM2.5 concentrations in Beijing by removing large 218 inter-annual and seasonal variations in meteorological conditions. The principle of the KZ 219 filter is briefly introduced as follows.

220 The raw time-series data of airborne pollutants can be decomposed as: (1) 221 X(t) = E(t) + S(t) + W(t) $X_b(t) = E(t) + S(t)$  (2) 222  $E(t) = KZ_{365,3}(X)$ (3) 223  $S(t) = KZ_{15,5}(X) - KZ_{365,3}(X)$ - (4) 224  $W(t) = X(t) - KZ_{15,5}(X)$   $W(t) = X(t) - KZ_{15,5}(X)$ 225 (5) 226 Where X (t) is the original time series of airborne pollutants, E(t) is the long-term trend component, 227 S(t) is the seasonal variation component, W(t) is the residue or synoptic-scale (short-term 228 (synoptic-scale)) variationscomponent or residue, KZi, j(X) indicates a-KZ filtering on the original 229 dataset X with a moving wind size of *i* and *j* iterations. 230 Xb(t) stands for the base component, the sum of the long-term-trend component and seasonal 231 variation component, presenting steady trend variation. E(t) is mainly effected affected by 232 long-term anthropogenic emission and climate change. S(t) is mainly influenced by the 233 seasonal variation of emission factors and meteorological conditions. The residue W(t) is

234 caused by short-term and small-scale shifts of emissions and meteorological conditions.

The long-term trend component E(t) processed by KZ filtering still contains the influence of meteorological conditions, which can be removed by multiple regression models. Multiple linear relationships are established for the residue and baseline component respectively using strongly correlated meteorological factors strongly correlated with airborne pollutants.

239 We conducted correlation analysis examined correlations between seasonal PM2.5 240 concentrations in Beijing and a series of meteorological factors, including temperature, wind 241 speed, wind direction, precipitation, relative humidity, solar radiation, evaporation and air 242 pressure-. Due to limited space, detailed correlations between PM2.5 concentrations and 243 individual meteorological factors in Beijing are not presented here and readers can refer to 244 our-previous studies for more information (Chen et al., 2017; 2018). The correlation analysis 245 revealed that wind speed, relative humidity, temperature and solar radiation were strongly 246 and significantly correlated with PM2.5 concentrations in Beijing, which was consistent with the findings from previous other studies (Sun et al., 2013; Chen, Z., et al., 2017, 2018; Wang 247

|-------|-----|------------|
|       |     |            |
|       |     |            |

et al., 2018). Therefore, we further established multiple linear regression equations between
PM2.5 concentrations and wind speed, relative humidity, temperature and solar radiation as
follows.

$$W(t) = a_0 + \sum a_i \mathbf{w}_i(t) + \varepsilon_w(t)$$

(6)

(7)

(8)

$$X_{b}(t) = \mathbf{b}_{0} + \sum b_{i} x_{i}(t) + \varepsilon_{b}(t)$$

 $\varepsilon(t) = \varepsilon_w(t) + \varepsilon_h(t)$

254 Where  $w_i(t)_{\star}$  and  $\chi_{ii}(t)_{\star}$  stand for the different short-termsynoptic scale variations and baseline 255 component of the ith meteorological factor.  $\varepsilon_{av_{\lambda}}$  and  $\varepsilon_{b}_{\star}$  is the regression residue of the 256 short-termsynoptic-scale variations and baseline component.  $\varepsilon(t)_{\star}$  indicates the total residue, 257 including the short-term influence of local emission sources, meteorological influencesand 258 meteorological factors neglected during the regression process and other noises.

Next, KZ filtering is was conducted on the  $\varepsilon(t)$  for its long-term component  $\varepsilon_E(t)$ . After the variation of meteorological influences was filtered, the reconstructed time series of airborne pollutants  $X_{LT}(t)$  was calculated as the sum of  $\varepsilon_E(t)$  and the average value of E(t),  $\overline{E(t)}$ .

263

$$X_{LT}(t) = E(t) + e_E(t) \tag{9}$$

After KZ filtering, the relative contribution of meteorological conditions to  $\underline{PM_{2.5}}$  variations the variation in  $\underline{PM_{2.5}}$  concentrations can be calculated as follows:

266
$$P_{contrib} = \frac{K_{org} - K}{K_{org}} \times 100\%$$
(10)

267 Where *P*contrib is the relative contribution of meteorological conditions to the variation of PM2.5 268 concentrations variations in Beijing, *K*org is the variation slope of the original PM2.5 time series; K is 269 the variation slope of adjusted PM2.5 time series after with filtered influences from meteorological 270 variations<del> are removed</del>.

**271 3.2 WRF-CMAQ model**

272 We employed the-WRF-CMAQ-model for simulating the effects of emission-reduction

|-------|----------------|
|       |                |
|       |                |
|       |                |

|-------|-----|------|----|----|--|--|
|       |     |      |    |    |  |  |

|---------|-----|------------------|---------|----|
| 带格式的: " | 字体: | Times New Roman, | 10.5 磅, | 加粗 |
| 带格式的: " | 字体: | Times New Roman, | 10.5 磅, | 加粗 |

273 measures on the reduction decrease of PM2.5 concentrations. The WRF-CMAQ model includes three models: The middle-scale meteorology model (WRF), the source emission 274 275 model (SMOKE) (http://www.cmascenter.org/smoke/) and the community multiscale air 276 quality modeling system (CMAQ) (http://www.cmascenter.org/CMAQ). The center of the 277 CMAQ was set at coordinate 35°N, 110°E and a bi-directional nested technology was 278 employed, producing two layers of grids with a horizontal resolution of 36 km and 12 km 279 respectively. The first layer of grids with 36km resolution and 200×160 cells covered most 280 areas in East Asia (including China, Japan, North Korea, South Korea, and other countries). 281 The second layer of grids with 12km resolution and 120×102 cells covered the North China 282 Plain (including the Beijing-Tianjin-Hebei region, and Shandong and Henan Provinces). The 283 vertical layer was divided into 20 unequal layers, eight of which were of a less-than-1km 284 distance of less than 1km to the ground for better featuring the structure of atmospheric boundary. The height of the ground layer was 35m. 285

286 We employed ARW-WRF3.2 to simulate the meteorological field. The setting of the center 287 and the bi-directional nest for the WRF and was similar to that of the CMAQ was similaras 288 mentioned above. There were 35 vertical layers for the-WRF and the outer layer provided 289 boundary conditions of the inner layer. The meteorological background field and boundary 290 information with a FNL resolution of 1°×1° and temporal resolution of 6h were acquired 291 from NCAR (National Center for Atmospheric Research, https://ncar.ucar.edu/) and NCEP 292 (National Centers for Environmental Prediction) respectively. The terrain and underlying 293 surface information was obtained from the USGS 30s global DEM 294 (https://earthquake.usgs.gov/). The outputs from-the WRF-model wereas interpolated to the 295 region and grid of the CMAQ-model using the Meteorology-Chemistry Interface Processor 296 (MCIP, https://www.cmascenter.org/mcip). The meteorological factors used for this model 297 includeds temperature, air pressure, humidity, geopotential height, zonal wind, meridional 298 wind, precipitat

---

## Author Response (AR2)

Comments to the Author:

You have not adequately addressed the comments made by the reviewers.

To be able to have this manuscript published I would like you to take into consideration the following remarks:

**To Editor:**

Thanks so much for providing us a chance to revise this manuscript again. We have carefully checked this manuscript and revised it fully according to your comments. Please feel free to contact us if further revisions are required.

In the Introduction, please make sure that you site all the relevant work that has been devoted to looking at trends of  $PM_{2.5}$  over Beijing. In addition, the last 3-4 sentences of the introduction should summarize the sections that you develop in the paper.

R: Thanks so much for this point. We have added more relevant works concerning trends of PM2.5 over Beijing. Meanwhile, the introduction section has been revised according to your suggestions.

We have carefully searched relevant publications that looking at trends of PM2.5 variations from 2013 to 2017, which is a specific period for evaluation. Since the completion of this period just passed for one year, not many relevant papers found. In the revised manuscript, we included another five papers that mentioned PM2.5 variations in Beijing from 2013 to 2017. These papers mainly discussed the spatial-temporal variations of PM2.5 variations in Beijing from 2013 to 2017. These papers mainly discussed the spatial-temporal variations of PM2.5 variations in Beijing from 2013 to 2017. Several of them employed some field-collected PM2.5 sample to analyze the source of PM2.5 component during short-term pollution episodes. Therefore, they are not highly correlated with the major aim of this research, the meteorological influences on PM2.5 variations from 2013 to 2017.

Shao, P., Tian, H., Sun, Y., Liu, H., Wu, B., Liu, S., Liu, X., Wu, Y., Liang, W., Wang, Y., Gao, J., Xue, Y., Bai, X., Liu, W., Lin, S., Hu, G.: Characterizing remarkable changes of severe haze events and chemical compositions in multi-size airborne particles (PM1, PM2.5 and PM10) from January 2013 to 2016–2017 winter in Beijing, China. Atmospheric environment, 189, 133-144, 2018.

Xu, H, Xiao Z, Chen K, Tang M, Zheng N, Li P, Yang N, Yang W, Deng X.: Spatial and temporal distribution, chemical characteristics, and sources of ambient particulate matter in the

Beijing-Tianjin-Hebei region. Science of The Total Environment, 658, 280-293, 2019.

Wang T, Du Z, Tan T, Xu N, Hu M, Hu J, Guo S.: Measurement of aerosol optical properties and their potential source origin in urban Beijing from 2013-2017. Atmospheric Environment, 206, 293-302, 2019.

Liang, L., Cai, Y., Barratt, B., Lyu, B., Chan, Q., Hansell, A.L., Xie, W., Zhang, D., Kelly, F.J., Tong, Z.: Associations between daily air quality and hospitalisations for acute exacerbation of chronic obstructive pulmonary disease in Beijing, 2013–17: an ecological analysis. The Lancet Planetary Health, 3(6), 270-279, 2019.

Sun, J., Gong, J., Zhou, J., Liu, J., Liang, J..: Analysis of PM2.5 pollution episodes in Beijing from 2014 to 2017: Classification, interannual variations and associations with meteorological features. Atmospheric Environment, 213, 384-394, 2019.

Zhai S, Jacob D J, Wang X, Shen L, Li K, Zhang Y, Gui K, Zhao T, Liao H.: Fine particulate matter (PM 2.5) trends in China, 2013–2018: separating contributions from anthropogenic emissions and meteorology. Atmospheric Chemistry and Physics, 19(16), 11031-11041,2019.

So we gave a general introduction of these studies concerning the specific trends of  $PM_{2.5}$  over Beijing during the Clean Air Action period

"The notable decrease of PM2.5 concentrations attracted nationwide attentions and growing studies have been conducted to understand spatio-temporal characteristics (Shao et al., 2018; Sun et al., 2019; Wang et al., 2019), sources (Chen et al., 2019; Xu et al., 2019; Cheng et al., 2019) and health effects (Liang et al., 2019) of PM2.5 variations in Beijing from 2013 to 2017. These studies revealed that air quality in Beijing was improved significantly in 2017 in terms of annual mean PM2.5 concentrations, polluted days and pollution durations. Furthermore, despite different outputs, both source apportionment during pollution episodes based on collected samples (Shao et al., 2019; Xu et al., 2019; Chen et al., 2019) and long-term model simulation based on regional and local emission inventories (Cheng et al., 2019) suggested that local and regional anthropogenic emissions (e.g. coal combustion and vehicle emissions) were the major influencing factors for long-term and short-term PM2.5 variations in Beijing."

We also added some introduction for a recently published paper concerning PM2.5-meteorology relationship across China from 2013 to 2018

"Based on a stepwise multiple linear regression (MLR) model, Zhai et al. (2019) quantified the relative contribution of meteorology to PM2.5 variations from 2013 to 2018 in Beijing-Tianjin-Hebei region, Yangtze River Delta, Pearl River Delta and Sichuan Basin and Fenwei plain was 14%, 3%, 19%, 27% and 23% respectively."

According to your comment, we added a short introduction of the structure of this manuscript at the end of the introduction section as follows:

"This manuscript is structured as follows: Firstly, major data sources, including PM2.5 and meteorological data, and emission inventories, employed for this research are briefly introduced. Secondly, the principle and parameter setting of two models, KZ filtering and WRF-CMAQ, and model verification are explained. In the result section, the relative contribution of meteorological conditions and anthropogenic emissions to PM2.5 variations in Beijing from 2013 to 2017 calculated using both models is presented. In the discussion and conclusion part, implementations of this research and suggestions for further improving air quality in Beijing are given."

**Thanks again for this valuable comments.**

Remarks to be addressed concerning reviewer 1:

I propose that you include the following Table with the relevant explanations from your answers to comment 3 from Reviewer 1 in the manuscript:

The comparison of The local environmental statistical data used for this research and other official statistical data in 2017 (unit: 10k tons) SO2 NOx CO VOC NH3 PM10 PM2.5 BC OC Statistical data for this research 1.38 10.15 49.54 13.47 3.20 14.74 3.92 0.17 0.44 National Environmental Statistics Bulletin 1.38 12.16 52.03 24.24 3.26 14.68 3.91 0.22 0.41 "2+26" center for air pollution prevention and control 0.89 9.24 48.98 13.93 3.16 13.82 3.72 0.19 0.46

R: Thanks so much for this comment. We have added this table and the following text to the revised manuscript.

"As shown in table 1, it is highly consistent with other official statistical data, such as the Annual report from National Environmental Statistics Bulletin (http://www.mee.gov.cn/gzfw 13107/hjtj/qghjtjgb/) and "2+26" Center for Air Pollution Prevention and Control, and has been formally employed for the implementation of recent "2017 Air Pollution Prevention and Management Plan for the Beijing-Tianjin-Hebei Region and its Surrounding Areas" (MEP, 2017)."

Where in the revised version do you indicate the following results that appears in your answer to Comment 4 of reviewer 1?

"This means the extracted seasonal component and short-term component made a significant contribution to seasonal and short-term variations of original PM2.5 concentrations in Beijing from 2013 to 2017, indicating a satisfactory KZ filtering result. »

R: Thanks so much for pointing this out. There are one major approach to verify the efficiency of KZ. If the total variations of long-term, seasonal and short-term component was close to 1, it suggests that a majority of meteorological influences has been considered and effectively removed. Specifically, the variation of seasonal (ranging from 9%-23.8%) and short-term component (ranging from 66.8%-83.8%) was much larger than that of long-term component (ranging from 1.2%-3.5%).

So in the revised manuscript, we included the following text

"The sum of the long-term, seasonal and short-term component contributed to more than 93.6~95.3% of the total variance in different stations respectively. The larger the total variance, the three components are more independent to each other. The total variance close to 100% suggests that a majority of meteorological influences has been considered and effectively removed. As shown in Table 3, the large value of the total variation in all stations indicated a satisfactory output from the KZ filtering.

Specifically, the relative contribution of the seasonal component (ranging from 9%-23.8%) and short-term component (ranging from 66.8%-83.8%) was much larger than that of the long-term component (ranging from 1.2%-3.5%), suggesting that seasonal and short-term variations of meteorological and emission factors exerted a major influence on the rapid change of PM2.5 concentrations in Beijing."

In addition to the statistical results, according to the comment 4 from reviewer 1, we added a Figure 2 to present the decomposed long-term, seasonal and short-term components using KZ filter. According to Figure 2, we can see that the long-term component demonstrates a smooth curve whilst the trend of season component and short-term component is highly consistent with that of the original PM2.5 time series, especially for some simultaneous peaks. Therefore, the seasonal and short-term variations of PM2.5 concentrations were effectively extracted as indicative seasonal component and short-term component. In the revised manuscript, we employed the following text to explain this

"According to Fig 2, the notable peaks of decomposed seasonal and short-term component were highly consistent with the peaks of PM2.5 concentrations in the original time-series, which further proved the dominant influence of seasonal and short-term variations of meteorological and anthropogenic factors on the temporal changes of PM2.5 concentrations in Beijing."

**You do not answer adequately the comment 5 of reviewer 1.**

"Q5. Section 3.2.2. Model evaluation is the key point in this paper. If the model data is not consistent with observation, contribution of emission control is out of the question. It seems that lots of data are far from the observation especially during the heavy air pollution days. So it is better to convert Fig 2 to time series plots, which can tell us more detailed information about the model evaluation."

Please draft an adequate answer and modify the manuscript accordingly.

R: Thanks so much for pointing this out. According to this comment, firstly, we have converted this Figure (Fig 3 in revised manuscript) to time series plots and included other three urban stations, which presented detailed information about the model evaluation.

As we acknowledged in the revised manuscript

"According to Fig 3, the general trend of the simulated PM2.5 concentrations was consistent with that of the observed PM2.5 concentrations. For six stations, the correlation coefficient R, normalized mean bias (NMB), normalized mean error (NME), mean fractional bias (MFB) and mean fractional error (MFE) between observed and simulated data was 0.63~0.91, -6%~6%, 26%~40%, -5%~7%, and 27%~46% respectively, indicating a satisfactory simulation output (EPA, 2005; Boylan et al., 2006). However, as shown in Figure 3, WRF-CMAQ may notably underestimate PM2.5 concentrations during heavy pollution episodes due to unified parameter setting for long-term simulation, the uncertainty in emission inventories, and especially insufficient chemical reaction mechanisms, which is a common challenge for CTM-based PM2.5 simulation (Li et al., 2011)."

The general accuracy of model simulation was satisfactory in terms of R, NMB, NME, MFB and MFE. Meanwhile, the long-term trend of simulated PM2.5 concentrations was consistent with that of observed PM2.5 concentrations. However, as the reviewer pointed out, WRF-CAMQ could lead to large variations during heavy pollution episodes, especially for long-term simulation with unified parameters (Li et al., 2011). We explained some underlying reasons for this common and unsolved challenge, the uncertainty in emission inventories, and especially insufficient chemical reaction mechanisms. We gave an example of this issue and its potential solution in the revised manuscript. More finer-scale emission inventories and better descriptions of reaction mechanisms in WRF-CAMQ can further improve simulation accuracy.

"For instance, without considering heterogeneous/aqueous reactions between multiple precursors, CTMs failed to approach the maximum PM2.5 concentrations during severe haze episodes and the simulation accuracy was dramatically improved by including proper descriptions of heterogeneous/aqueous reactions into CTMs (Chen, D. et al. 2016). With more finer-scale emission inventories and better descriptions of reaction mechanisms between precursors, the accuracy of PM2.5 simulation can be improved significantly"

In your revised manuscript 2 Tables are referenced as Table 3. You should have picked up this mistake.

R: We are very sorry for this mistake. We have checked the manuscript carefully and revised this and other typos.

In the new Figure 3 you have 6 plots of timeseries and 3 scatterplots. I want a descriptive Figure caption for ALL of them.

In the new Figure 4 you have 6 plots of timeseries and 2 plots with stacked lines. Please write a Figure caption for ALL of them.

R: I think there is some misunderstanding here. It may be attributed to the change-track version of the manuscript and deleted figures may appeared as part of the new Figures. Actually, in the clean version of the revised manuscript, according to the reviewer 1's comments, for Figure 3, we simply have 6 plots of time series images. As follows:

Fig 3. The comparison between observed and WRF-CMAQ simulated PM2.5 concentrations in 2017 in six stations across Beijing

For figure 4, we simply have 6 plots of time series. As follows: